# Proteomic analysis of SARS-CoV-2 particles unveils a key role of G3BP proteins in viral assembly

Emilie Murigneux[1,7], Laurent Softic[1,7], Corentin Aubé[1,7], Carmen Grandi [2], Delphine Judith[1], Johanna Bruce [3], Morgane Le Gall [3], François Guillonneau [3,6], Alain Schmitt [1], Vincent Parissi [4], Clarisse Berlioz-Torrent[1], Laurent Meertens[5], Maike M. K. Hansen [2] & Sarah Gallois-Montbrun [1]✉

Considerable progress has been made in understanding the molecular host-virus battlefield during SARS-CoV-2 infection. Nevertheless, the assembly and egress of newly formed virions are less understood. To identify host proteins involved in viral morphogenesis, we characterize the proteome of SARS-CoV-2 virions produced from A549-ACE2 and Calu-3 cells, isolated via ultracentrifugation on sucrose cushion or by ACE-2 affinity capture. Bioinformatic analysis unveils 92 SARS-CoV-2 virion-associated host factors, providing a valuable resource to better understand the molecular environment of virion production. We reveal that G3BP1 and G3BP2 (G3BP1/2), two major stress granule nucleators, are embedded within virions and unexpectedly favor virion production. Furthermore, we show that G3BP1/2 participate in the formation of cytoplasmic membrane vesicles, that are likely virion assembly sites, consistent with a proviral role of G3BP1/2 in SARS-CoV-2 dissemination. Altogether, these findings provide new insights into host factors required for SARS-CoV-2 assembly with potential implications for future therapeutic targeting.

The Severe Acute Respiratory Syndrome Coronavirus 2 (SARS-CoV-2) is the causal agent of the Coronavirus Disease 2019 (COVID-19) pandemic that emerged in late 2019[1]. Impressively, rapidly developed anti-COVID-19 vaccines have limited the evolution of symptoms towards severe pathologies. However, there is currently no prophylactic treatment available, and a growing number of studies suggest that the virus can persist in certain tissues in some patients over a long period of time, possibly contributing to long-COVID[2]. To control the emergence of more contagious strains, a better understanding of the molecular mechanisms involved in SARS-CoV-2 replication is needed.

After SARS-CoV-2 entry into host cells, viral genomic RNA (gRNA) is directly translated in the cytoplasm into two large polyproteins that are subsequently cleaved to generate non-structural proteins (nsps). These nsps assemble and recruit host cell proteins to form a replication and transcription complex, localized in a virus-induced network composed of double-membrane vesicles (DMVs)[3–5]. Within DMVs, the RNA-dependent RNA polymerase uses gRNA as a template for synthesis, via negative-strand intermediates, of both progeny genomic RNA (gRNA) and subgenomic RNAs (sgRNAs). Newly synthetized RNAs are then exported through transmembrane pores of DMVs to be

[1]Université Paris Cité, CNRS, Inserm, Institut Cochin, F-75014 Paris, France. [2]Institute for Molecules and Materials, Radboud University, 6525 AJ Nijmegen, the Netherlands. [3]Proteom'IC facility, Université Paris Cité, CNRS, Inserm, Institut Cochin, F-75014 Paris, France. [4]Microbiologie Fondamentale et Pathogénicité Laboratory (MFP), UMR 5234, « Mobility of pathogenic genomes and chromatin dynamics » team (MobilVIR), CNRS-University of Bordeaux, DyNAVIR network, Bordeaux, France. [5]Université Paris Cité, Inserm U944, CNRS 7212, Institut de Recherche Saint-Louis, Hôpital Saint-Louis, Paris, France. [6]Present address: Institut de Cancérologie de l'Ouest (ICO), CRCi2NA-Inserm UMR 1307, CNRS UMR 6075, Nantes Université, Angers, France. [7]These authors contributed equally: Emilie Murigneux, Laurent Softic, Corentin Aubé. ✉e-mail: sarah.gallois-montbrun@inserm.fr

subsequently used for translation of viral proteins. The gRNA is also packaged into new viral particles[6]. The viral N protein plays a strategic role in genome packaging by assembling as a succession of ribonucleoproteins (RNPs) along the genomic RNA in a bead on a string conformation[4,7]. Interestingly, N can phase separate in the presence of RNA and it was proposed that liquid-liquid phase separation may favor the condensation of viral RNPs and the packaging of the unusually large SARS-CoV-2 gRNA into virions[8–13]. Viral RNPs are then recruited by the M protein and assemble with the other structural proteins at modified cellular membranes derived from the endoplasmic reticulum (ER), the Golgi, the reticulum-Golgi intermediate compartment (ERGIC), and in surrounding single membrane vesicles (SMVs). Finally, newly formed virions are released into SMVs and exported to the plasma membrane[3,6,7,14,15].

Like other RNA viruses, the SARS-CoV-2 genome encodes a limited number of viral proteins and relies heavily on cellular machinery for production of virions and subsequent infection[16,17]. To delineate host-virus interactions taking place during viral replication, numerous large-scale screens have been performed. Examination of the effect of SARS-CoV-2 infection on the cellular proteome revealed that the virus reshapes expression of cell host factors to accommodate and optimize its replication[18]. Genome-wide CRISPR-Cas9 screens were conducted to recover host factors involved in the resistance (antiviral), or sensitization (pro-viral) of cells to the cytopathic effect of SARS-CoV-2[1,19–24]. Surprisingly, minimum overlap in hit identification was obtained across these studies. Furthermore, identified proteins were mainly involved in early steps of viral replication, and the late stages of viral particle assembly and release were less well explored. Examination of host factors interacting with viral proteins[25–28], as well as with viral RNAs[29–36] also highlighted that hundreds of factors, including numerous RNA binding proteins (RBP) are involved in the virus-host cell battlefield[37]. Nevertheless, detailed mechanistic understanding of these interplays at late stages of infection remains limited and none of the above studies specifically interrogated factors involved in gRNA packaging, particle assembly and egress[16,17].

As for most enveloped viruses, host encoded proteins are susceptible to be packaged within SARS-CoV-2 virions during viral morphogenesis[38]. With the underlying idea that host proteins important for viral particle assembly are likely required at the assembly site and may be incorporated into virions during this process, we set out to identify cellular factors associated with SARS-CoV-2 virions. Through two complementary approaches, we isolated virions produced from two pulmonary epithelial cell lines. Liquid chromatography-coupled mass spectrometry (LC-MS) analyses resulted in the identification of 356 putative host factors associated with SARS-CoV-2 virions. In particular, we highlighted the presence of a set of RBPs that can localize to cytoplasmic membraneless structures called stress granules (SGs). G3BP1 and G3BP2 (G3BP1/2), two GTPase-activating protein (SH3 domain) binding proteins, play a central role in SG formation and maintenance[39–43]. In view of their incorporation into SARS-CoV-2 virions, we assessed the contribution of G3BP1/2 to different steps of viral replication. Our study uncovers an unexpected role of G3BP1/2 in the formation of cytoplasmic membrane structures that concentrate N, S, M proteins and gRNA to promote production of infectious SARS-CoV-2 virions.

## Results

### Capturing proteins associated with SARS-CoV-2 virions
To identify host cell proteins potentially present in extracellular SARS-CoV-2 virions, viral particles were produced from two lung epithelial cell models, A549 cells overexpressing ACE2 receptor (A549-ACE2) and Calu-3 cells that naturally express ACE2 receptor and TMPRSS2. Following infection with the D614G SARS-CoV-2 strain[32], culture supernatants were collected between 40 and 48 h post-infection (hpi), before cells exhibited any visible virus-induced cytopathic effects, and

pelleted over a 20% sucrose cushion. An ultrafiltration step was included to further eliminate contaminating proteins coming from lysed cells and the culture media. As a control, supernatants of mock-infected cells were similarly ultracentrifuged and contents of both preparations were compared (Fig. 1a).

Western blot and qPCR analysis showed that the viral structural proteins N, M, cleaved (S2) and uncleaved (S0) spike glycoproteins, as well as viral gRNA were exclusively detected in virion preparations (Fig. 1b, c and Supplementary Fig. 1a). Moreover, viral subgenomic N RNA (N sgRNA), that is highly expressed in infected cells was not detected by qPCR, indicating limited cellular contaminations of viral preparations (Supplementary Fig. 1b). We also checked the infectivity of isolated viruses using plaque assay (Fig. 1d), and confirmed by transmission electron microscopy a large enrichment of intact single and clustered virions with an 80 to 100 nm diameter and with typical corona-like features (Fig. 1e)[44].

Ultracentrifuged supernatants from mock- and SARS-CoV-2-infected cells were then subjected to LC-MS analysis. The resulting peptides were used to interrogate the SARS-CoV-2 database that includes all viral proteins. N, S and M proteins were highly enriched in isolated virions from both cell lines, as compared to supernatants of mock-infected cells (Fig. 1f, g, blue dots). Given its small size (8 kDa), E protein only yielded 1–3 tryptic peptides and was not detected in the viral preparations. ORF7a, a SARS-CoV-2 accessory protein likely involved in counteracting virion retention by BST2, was enriched, although to a lesser extent than the structural proteins[45] (Fig. 1b, f (blue dot), Supplementary Fig. 1a, c). Importantly, other nsps were not detected, even those that are more highly expressed in cells than S proteins, such as nsp3, consistent with little or no contaminant of cellular components in purified virion preparations (Fig. 1b, Supplementary Fig. 1a, c and Supplementary Data 1)[46].

To identify host factors associated with virion, LC-MS data was then used to query a human protein database. In total, 358 cellular factors were identified in at least 3 biological replicates in preparations from non-infected and infected A549-ACE2 cells (Fig. 1f) of which 284 proteins were statistically enriched in SARS-CoV-2 preparations compared to mock-infected controls with a fold change FC ≥ 1.5 (adjusted $p \le 0.05$) (in orange, Fig. 1f, Supplementary Data 1). Preparations from Calu-3 cells identified 943 host factors from 3 biological replicates (Fig. 1g). However, a significant overlap in protein identification was observed between preparation from non-infected and infected cells. This could be attributed to greater extracellular vesicles and/or mucus production inherent to Calu-3 cells[47], leading to a higher background as compared to A549-ACE2 samples. Nonetheless, 120 proteins were statistically enriched in virions ($p \le 0.05$, FC ≥ 1.5). This included 48 proteins that overlap with proteins identified in virions produced from A549-ACE2 cells, underscoring their significance in shaping the composition of SARS-CoV-2 virions (Fig. 1g). Overall, a set of 356 human proteins were identified as virion-associated factors from both cell lines (Supplementary Data 1).

### Cellular factors identified with SARS-CoV-2 virions interact with each other and with viral structural proteins and/or viral RNA in infected cells
To examine the possible connectivity of SARS-COV-2 virion-associated factors, we first built a protein-protein association network based on known interactions within cells using STRING software[48]. Virion-associated factors derived from A549-ACE2 cells are represented by ellipses, those from Calu-3 cells by diamonds and those common to both cell lines by polygons (Fig. 2a). Eighty-three percent (297/356) of virion-associated proteins belonged to a highly reticulated network, each protein interacting on average with 12.1 neighbors, which is higher than a random distribution within a control network of the same size ($p < 1 \times 10^{-16}$) (Fig. 2a). Even after removing ribosomal factors from the analysis, which may artificially increase the number of

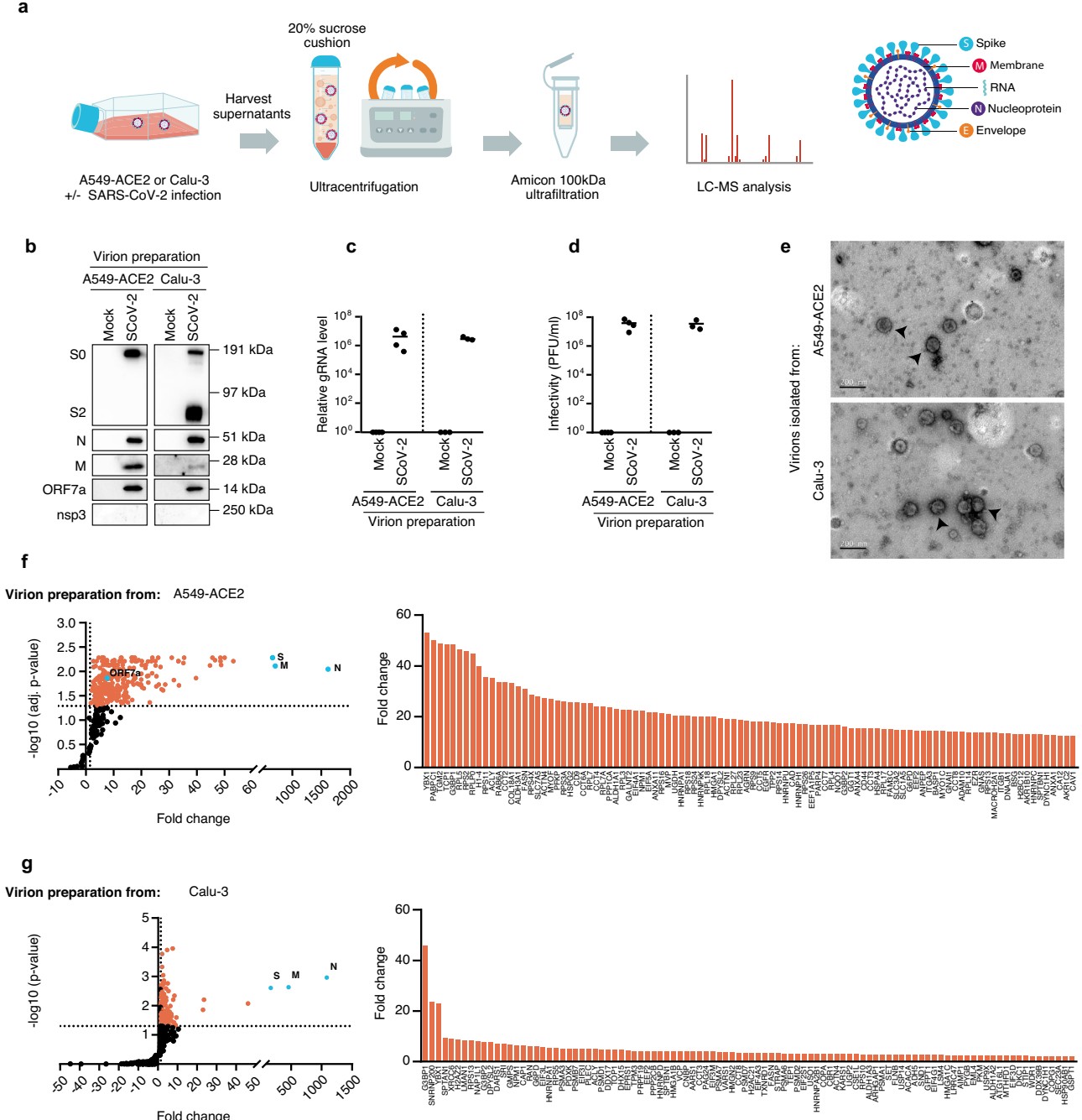

**Fig. 1 | Proteomic analysis of factors associated with SARS-CoV-2 virions isolated on sucrose cushion and ultrafiltration. a** Outline of SARS-CoV-2 virion preparation for LC-MS analysis: A549-ACE2 and Calu-3 cells were either mock- or SARS-CoV-2-infected for 48 and 40 h, respectively. Supernatants were harvested and subjected to ultracentrifugation on a 20% sucrose cushion. Isolated virions were lysed and analyzed by LC-MS. A schematic representation of a SARS-CoV-2 virion highlighting the structural proteins is shown. **b** Representative immunoblotting of uncleaved S0 and cleaved S2, N, M, ORF7a and nsp3 viral proteins in virion preparations using the indicated antibodies. For analysis, 1.25% of the input, flow through and affinity capture samples were loaded. **c** Quantification of SARS-CoV-2 genomic RNA (gRNA) by qPCR in virion preparations from mock- or SARS-CoV-2-infected A549-ACE2 and Calu-3 cells ($n$ = 4 and $n$ = 3 independent experiments, respectively). **d** Infectious viral titer of virion preparations derived from A549-ACE2 and Calu-3 cells ($n$ = 4 and $n$ = 3 independent experiments,

respectively), quantified by plaque assay. **e** The presence of intact virions was assessed by electron microscopy in virion preparations from SARS-CoV-2 infected A549-ACE2 cells and Calu-3 cells. Black arrows indicate typical SARS-CoV-2 virions in representative images of two independent experiments, scale bar = 200 nm. Cellular factors identified by LC-MS analysis in preparations from mock- and SARS-CoV-2-infected A549-ACE2 (**f**) and Calu-3 (**g**) cells. Scatter plots show statistically enriched SARS-CoV-2 viral proteins (blue dots) and host factors (orange dots) in virion preparations with (**f**) adj. $p \leq 0.05$ and FC ≥ 1.5, ($n$ = 4 independent experiments) and (**g**) $p \leq 0.05$, FC ≥ 1.5, ($n$ = 3 independent experiments). One-tailed paired Student's $t$-test adjusted with Benjamini–Hochberg correction and one-tailed paired Student's $t$-test were applied in **f** and **g**, respectively. The mean fold change of the top one hundred virion-associated factors is detailed. Source data are provided as a Source Data file.

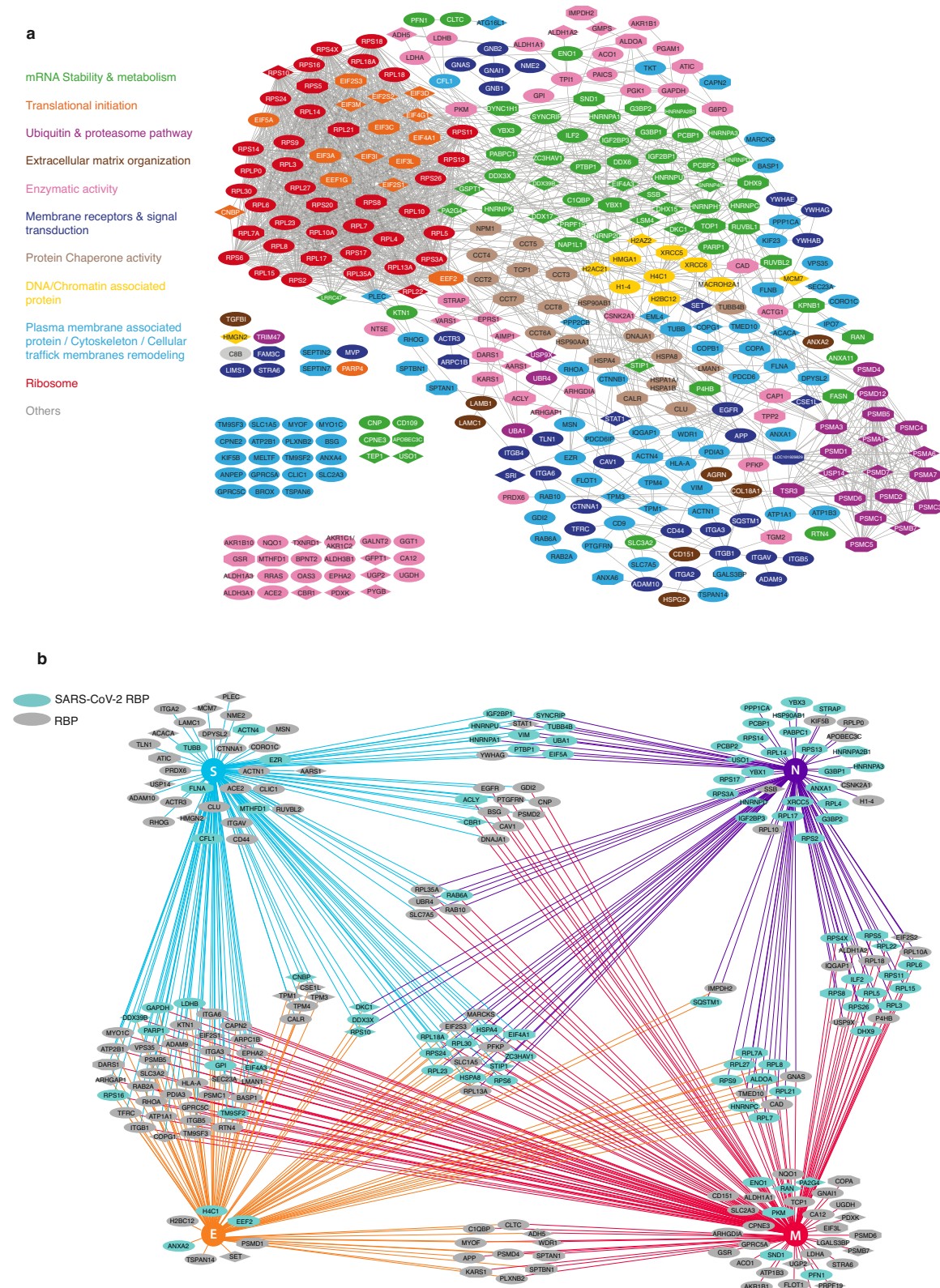

interactants, each protein still interacted on average with 4.77 other factors ($p < 1 \times 10^{-16}$), indicating that proteins associated with SARS-CoV-2 virions produced by A549-ACE2 and Calu-3 cells, are physically closely connected within the cells.

To gain insight into their biological functions, KEGG pathway and Gene Ontology (GO) enrichment analyses were performed (Supplementary Fig. 2a). KEGG analysis indicated that virion-associated factors

were highly enriched in COVID-19 disease related pathways, as well as ribosome and proteasome pathways, that are also strongly related to SARS infection[49,50] (Supplementary Fig. 2a). In line with this, we noticed a large number of ribosomal proteins and translation initiation factors (in red and orange, Fig. 2a), as well as a number of proteasomal sub-units (in purple, Fig. 2a). GO analysis confirmed a strong enrichment for terms linked to viral and mRNA metabolic processes (translation,

**Fig. 2 | Host factors associated with SARS-CoV-2 virions interact with each other and with viral RNA and structural proteins within infected cells.**
**a** Protein-Protein interaction network of host factors associated with virions produced from A549-ACE2 and Calu-3 cells was built based on curated interactions in STRING. The nodes represent identified host proteins associated with virions from A549-ACE2 cells (ellipses) and Calu-3 cells (diamonds) or both (polygons) and the edges represent known cellular interactions between two proteins of the network. Proteins without connection are shown on the left. Proteins were sorted according to their cellular and molecular functions and are highlighted in different colors

accordingly. Detailed KEGG and GO term analyses are provided in Supplementary Fig. 2a. **b** Protein-protein interaction network of virion-associated factors with SARS-CoV-2 structural proteins was built based on known interactions within infected cells (BIOGRID-PROJECT-covid19). Edges represent direct interactions of cellular proteins with one of the viral structural proteins, N (purple), M (red), S (blue) and E (orange). Proteins in cyan were previously shown to interact with SARS-CoV-2 RNA in the cells[29–33,54]. Protein in grey are RNA binding proteins (RBP) (according to RBPase v0.2.1 alpha).

transport, stability) as well as RNA binding amongst the molecular functions (in green, Fig. 2a, Supplementary Fig. 2a). Numerous membrane proteins, including receptors and proteins involved in signal transduction pathways (in dark blue, Fig. 2a), as well as proteins involved in membrane remodeling, cell trafficking and cytoskeleton were identified (in light blue, Fig. 2a), testifying to the large cellular membrane reorganization required to produce SARS-CoV-2 virions. Other proteins with chaperone and enzymatic activities were also identified (Fig. 2a, in beige and pink, respectively). Of note, tetraspanins were identified in both mock- and infected-A549-ACE2 and Calu-3 derived samples. However, classical exosome markers, CD63, CD81, and TSG101, were not enriched in virion preparations, suggesting limited exosome contaminants among identified virion-associated factors (Supplementary Fig. 1a, c and Supplementary Data 1).

Cross-referencing our data with a previously published transcriptomic study performed in nasopharyngeal swab and lung biopsies of patients infected with SARS-CoV-2[51] further indicated that 97.7% (348/356) of SARS-CoV-2 virion-associated factors were detected in natural cellular targets of infection (Supplementary Fig. 2b).

Comparison of the composition of SARS-CoV-2 virion-associated proteins with that of 20 other enveloped viruses revealed a markedly distinct profile[38,52]. However, 67% (240/356) of SARS-CoV-2 associated proteins had prior associations with other viruses (in blue, Supplementary Fig. 2c). These include host proteins like APOBEC3C, ZAP, and multiple ribosomal proteins, implying their specific inclusion in virion compositions. Additionally, we identified 116 proteins, such as G3BP1 and G3BP2, not typically found in other virions, suggesting a unique role for these proteins in SARS-CoV-2 infection (in yellow, Supplementary Fig. 2c).

Virion assembly inherently incorporates a portion of the cytoplasm into the viral particles, potentially resulting in the passive inclusion of abundant cellular proteins. In contrast, we anticipate that host factors actively engage with virions by interacting with specific virion components. To further investigate whether identified factors interact with structural components of the SARS-CoV-2 virion within host cells, we exploited BIOGRID-PROJECT-covid19 v4.4.221 that collates protein-protein interactions of each of the 4 structural proteins identified by various technologies[53]. Sixty six percent (236/356) of virion-associated factors could interact with at least one of the structural proteins in cells (Fig. 2b). In particular, 191 factors (54%) interacted with either N, M or both proteins that play key roles in gRNA packaging and virion assembly. Similar to other RNA viruses such as IAV or HIV-1, 98% of SARS-CoV-2 virion-associated factors were RBP. Moreover, intersecting our data with factors recently identified as SARS-CoV-2 RBP in different studies[29–33,54] revealed that 32.6% (116/356) of the virion-associated factors interacted with SARS-CoV-2 RNAs, strongly suggesting that these proteins are incorporated within the virion, rather than decorating the surface of the particle (Fig. 2b, blue nodes). In particular, 27 factors belonged to a core of 88 SARS-CoV-2 RBPs identified in common in at least three different studies (Supplementary Data 1). These proteins likely represent a specific subset of the 503 SARS-CoV-2 RBPs identified in the literature[29–33,54]. For example, viral proteins otherwise strongly associated with gRNA during replication, such as nsp3, nsp6 or nsp9[29,31,33,54], were not detected in virions (Fig. 1b, Supplementary Fig. 1a, c, Supplementary Data 1).

Although we cannot entirely exclude that some of these factors are passively associated with virions or enriched in extracellular vesicles that have incorporated the gRNA and/or one of the viral structural proteins, our data is consistent with an active recruitment of factors that are physically and functionally connected with each other and with one or several virion components during cell infection (Fig. 2).

## A subset of stress granule proteins is incorporated within SARS-CoV-2 virions

To further refine our analysis, we next captured SARS-CoV-2 virions produced from A549-ACE2 cells by affinity for its cellular receptor using biotinylated-ACE2 recombinant protein or biotin as a control in the presence of streptavidin coated beads (Fig. 3a). As expected in the biotinylated-ACE2 capture, N, S and ORF7a proteins, as well as gRNA were depleted from viral supernatant and enriched in the capture. In contrast, no enrichment was observed in the biotin control condition (Fig. 3b, c and Supplementary Data 1). Seventy cellular factors statistically enriched in ACE2 captures from 4 biological replicates were identified by LC-MS analysis ($p \leq 0.05$, FC ≥ 1.5) (Fig. 3d), including 57 proteins in common with factors enriched in the former viral preparations by sucrose cushion and ultrafiltration from A549-ACE2 cells (Supplementary Fig. 3a). None of these proteins were detected in ACE2 capture from non-infected supernatant in a control experiment (Supplementary Fig. 3b and Supplementary Data 1). As previously observed, these factors belonged to a highly reticulated interaction network within the cells. In the cellular context, most of them could interact with one or several viral structural proteins, in particular with N and M proteins, as well as with SARS-CoV-2 RNA (Supplementary Fig. 3c).

In total, 92 host factors associated with SARS-CoV-2 virions produced from A549-ACE2 cells and isolated via sucrose cushion and ultrafiltration were validated either in the viral preparation from Calu-3 cells and/or using ACE-2 affinity capture (Supplementary Fig. 3a), reinforcing their contribution to the composition of SARS-CoV-2 virion. Intriguingly, a third (26/92) of these factors belonged to dynamic mRNP granules called stress granules (SGs) (in purple, Supplementary Fig. 3a and Supplementary Data 1). SGs are usually formed by phase separation of RNAs and proteins as a result of translation shut-off in response to various cellular stresses, including viral infections[55,56]. Remarkably, SG proteins G3BP1, YBX1 and PABPC1 were amongst the proteins with the highest enrichment in virions produced from both A549-ACE2 and Calu-3 cells (Figs. 1f, g and 3d). Western-blot analyses further confirmed the association of G3BP1, G3BP2, YBX1, IGF2BP1, IGF2BP3 and PABPC1 with SARS-CoV-2 virions isolated by ultracentrifugation on sucrose cushion and ultrafiltration from both A549-ACE2 and Calu-3 cells, as well as by ACE2 affinity capture (Fig. 3e, f). Several other core SG factors such as TIA1, TIAR and CAPRIN were not identified with SARS-CoV-2 virion, neither in LC-MS analysis (Supplementary Data 1), nor by western-blot, despite their high expression levels in infected cells (Fig. 3e, f and Supplementary Fig. 3b). This suggests a specific incorporation of a subset of SG proteins into SARS-CoV-2 virions.

To further distinguish whether these virion-associated factors resided within viral particles or attached to their surfaces, isolated virions were subjected to subtilisin A treatment[57]. As expected, the full length 190 kDa spike protein (S0) was no longer detectable by

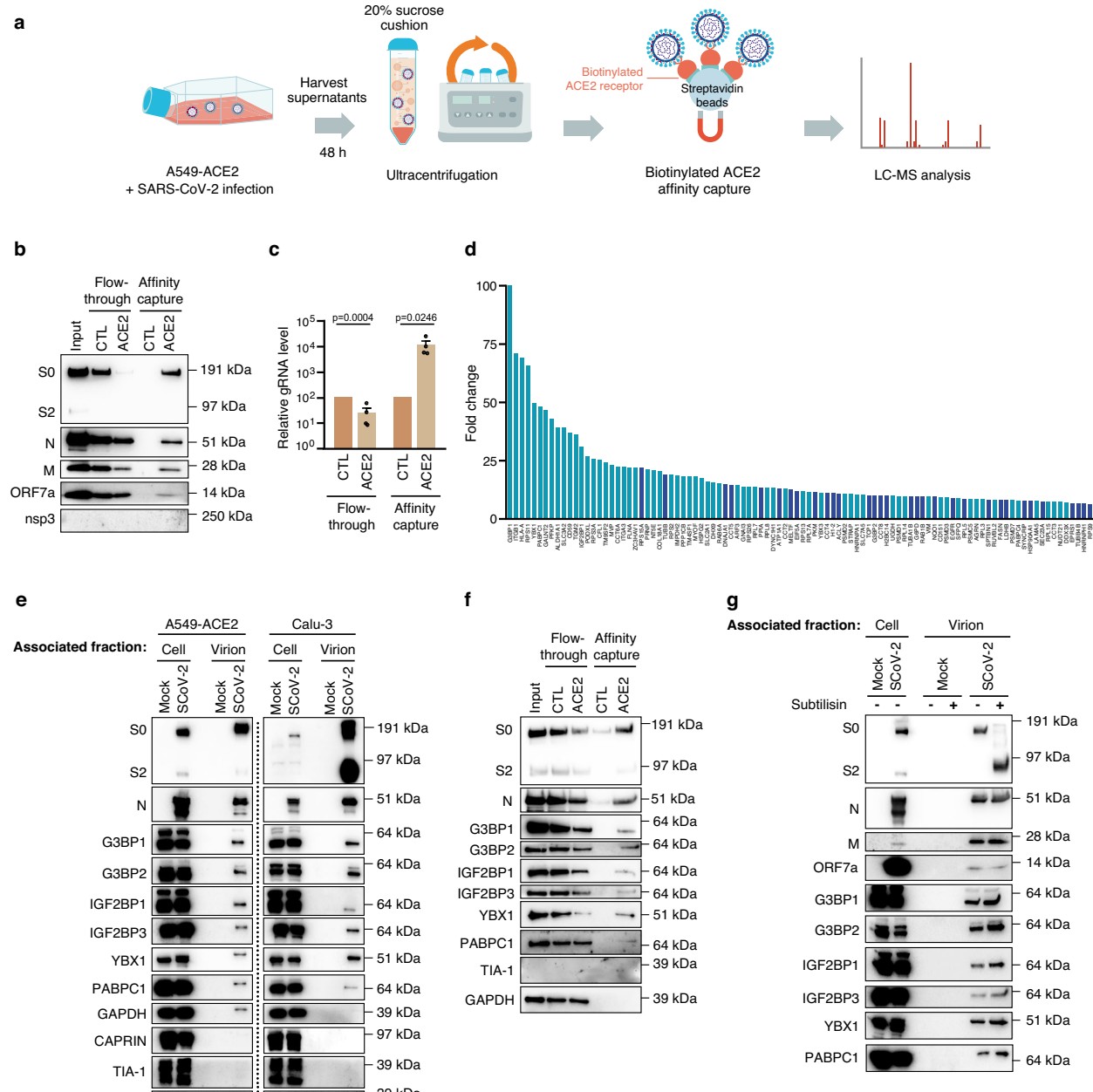

**Fig. 3 | Isolation of SARS-CoV-2 viral particles by affinity capture with ACE2 receptor identifies a subset of stress granule proteins associated with virions.**
**a** Outline of SARS-CoV-2 virions affinity capture with biotinylated ACE2 receptor. Virions produced from A549-ACE2 cells were isolated using ultracentrifugation on sucrose cushion (Input) as in Fig. 1a and further captured on streptavidin beads using either ACE2 receptor conjugated to biotin or biotin alone as a control.
**b** Uncleaved S0, cleaved S2 and N proteins were detected in input (as defined in a), flow-through and ACE2 affinity capture or biotin control by immunoblotting with the indicated antibodies. For analysis, 1.25% of the input, flow-through and affinity capture samples were loaded (representative image of 4 independent experiments). **c** Viral genomic RNA (gRNA) level in flow-through, ACE2 affinity capture and biotin control was quantified by qPCR. Data are presented as mean values +/− SEM ($n = 4$ independent experiments). One-tailed unpaired Student's $t$-test was applied. **d** The top one hundred cellular proteins, statistically enriched ($p \le 0.05$ in

light blue and $p \le 0.1$ in dark blue), in virions isolated on a sucrose cushion and purified using ACE2 affinity capture, are displayed based on their mean fold change as compared to biotin control ($n = 4$ independent experiments). One-tailed paired Student's $t$-test was applied. **e**–**g** Stress Granule (SG) proteins associated with SARS-CoV-2 virions were detected by immunoblotting with the indicated antibodies (images representative of $n = 2$ independent experiments). **e** For analysis, 0.25% of cell-associated fraction and 2% of virion isolated on sucrose cushion and ultra-filtration (as in Fig. 1a) from mock- or SARS-CoV-2-infected A549-ACE2 cells and Calu-3 cells were loaded. **f** For analysis, 1.25% of the input, flow through and virions isolated by ACE2 affinity capture or biotin control (as defined in a) from A549-ACE2 cells were loaded. **g** For analysis, 0.25% of cell-associated fraction and 2% of virion-associated fraction treated or not with subtilisin A were loaded. Source data are provided as a Source Data file.

immunoblotting. Only the C-terminal portion of the protein, migrating at 90 kDa and protected inside the virions from the serine protease digestion, was detected. In contrast, SG proteins associated with SARS-CoV-2 virion were still detected after subtilisin treatment, indicating their incorporation inside the viral particles (Fig. 3g).

## G3BP proteins favor SARS-CoV-2 replication at a late stage of infection
In most cases, induction of SG following viral infections is used by the cells as a host antiviral strategy. However, in some instances, viruses can exploit SG proteins to favor their replication[58–60]. In the context of

SARS-CoV-2 infection, the role of SG proteins has remained elusive[61–65]. To assess whether the incorporation of SG factors into SARS-CoV-2 virions is functionally relevant during replication, their expression was knocked-down (KD) by siRNA in A549-ACE2 cells. G3BP1 and G3BP2 being structurally and functionally related, the downregulation of G3BP1 led to an increase in the expression of G3BP2 and conversely (Supplementary Fig. 4a)[40,41,66]. We thus set out to downregulate the expression of both proteins simultaneously. A downregulation of more than 70% of RNA levels for all cellular candidates was confirmed (Supplementary Fig. 4b). Cells were then infected at a low multiplicity of infection (MOI 0.01) and viral replication was monitored over multiple rounds of infection by quantifying the level of viral gRNA in the virion-associated fraction at 48 hpi (Fig. 4a). KD of G3BP1/2 led to a 4.6-fold decrease in the level of virion-associated gRNA compared to control, suggesting a proviral role of these factors in SARS-CoV-2 replication. In line with their roles in SARS-CoV-2 mRNA stability and translation, KD of IGF2BP1 and PABPC1 also led to a 3.5- and 7.8-fold decrease, respectively[30,54]. By contrast, KD of YBX3 and IGF2BP3 had no impact, and KD of YBX1 increased the level of virion-associated gRNA by 3.1-fold, suggesting an antiviral role of this factor in the context of SARS-CoV-2. Overall, these data indicate that SG proteins associated with SARS-CoV-2 virions have different impacts on SARS-CoV-2 replication.

The role of G3BP proteins in the context of SARS-CoV-2 replication was controversial: whereas two studies indicated an antiviral function of G3BP1, two others suggested a proviral role[61–63,67]. To better delineate the function of these factors in viral replication, we further examined the impact of G3BP1/2 downregulation in a single round of infection. Following transfection with siRNA, A549-ACE2 cells were infected for 1 h at a high MOI (MOI 1), washed and harvested at 10 hpi together with the virion containing supernatant. Cell viability after siRNA treatments was assessed in parallel (Supplementary Fig. 4c). Whereas viral gRNA and N sgRNA levels produced in infected cells (cell-associated) were not affected by G3BP1/2 KD (Fig. 4b and Supplementary Fig. 4d), the level of gRNAs associated with virions in the supernatant (virion-associated) was decreased 2.2-fold (Fig. 4b). The SARS-CoV-2 release index, defined by the virion-associated gRNA level normalized to the cell-associated gRNA level, was decreased by 2-fold (Fig. 4b). Moreover, while the intracellular level of N protein quantified by immunoblotting was not affected by G3BP1/2 KD, suggesting that translation of viral proteins is not impacted at 10 hpi, the virion-associated N level was decreased by 1.8-fold, leading to a 1.7-fold decrease in the release index (Fig. 4c). This release index decrease was confirmed by ELISA quantification of N protein in both the cell and virion-associated fractions (Supplementary Fig. 4e). An even greater effect on the release index was observed for S protein with a 4.5-fold reduction, possibly related to an additional defect in S virion incorporation (Fig. 4c). Virion-associated fraction were then challenged in A549-ACE2 cells in a second round of infection, and a 3.6-fold decrease in viral titer (TCID50/mL) was observed in the virion-associated fraction of G3BP1/2 KD cells compared to the control (Fig. 4d). Viral infectivity, defined as the viral titer normalized to the level of virions assessed by N ELISA quantification was calculated, and revealed a 2-fold decrease (Fig. 4e). Similarly, in a population of transiently transduced A549-ACE2 cells with CRISPR-Cas9 targeting G3BP1 and G3BP2, the release index of gRNA, N and S proteins along with the viral titer and infectivity of the virion-associated fraction significantly decreased compared to the control population (Supplementary Fig. 4f–h). These findings were further validated in Calu-3 KD for G3BP1 and G3BP2 (Fig. 4b–e), further confirming the role of G3BP1/2 in producting infectious virions in different pulmonary cell models. Ectopic expression of G3BP2-HA in A549 cells resulted in a modest 1.2-fold increase in the release index, quantified at the RNA level. As expected, in a clone of A549 cells KO for both G3BP1 and G3BP2 (ΔG3BP1/2), the release index decreased by 1.6-fold as compared to control cells.

Importantly, G3BP2-HA overexpression in ΔG3BP1/2 cells led to a 3.4-fold increase compared to the ΔG3BP1/2 cells, significantly reversing the effect of G3BP1/2 knock-out (Fig. 4f).

Collectively, these data argue that G3BP1/2 favor the production and release of infectious virions in the supernatant at a post-translational level.

## G3BP proteins are recruited to sites of virion assembly and/or accumulation

To understand how G3BP1/2 favor viral particle production, we investigated the natural distribution of G3BP1 together with the viral structural proteins N, S and M during SARS-CoV-2 infection in the absence of SG-induced treatment. After multiple rounds of infection, a variable spatial distribution of N could be observed in cells, likely reflecting different stages of SARS-CoV-2 infection. N can form cytoplasmic puncta or present a more dispersed distribution characteristic of later stages of infection (Supplementary Fig. 5a). M and S were mostly present in perinuclear structures compatible with their localization in the Golgi apparatus (Supplementary Fig. 5a)[26,68]. In line with previous reports, the localization of N was thus globally distinct from the other structural proteins (Pearson < 0.2, Supplementary Fig. 5b)[26,68]. Interestingly, closer inspection of individual cells harboring a granular distribution of N revealed a striking accumulation of G3BP1 in some of N cytoplasmic puncta (Fig. 5a–d). Importantly, S and M also accumulated within some of these N/G3BP1 puncta, indicating that they are sites of viral structural protein accumulation (Fig. 5a, b). Although double strand RNA (dsRNA) staining, a presumed marker of viral RNA replication, was often in the vicinity of N/G3BP1 puncta, their colocalization was a rare event that only occurs in dsRNA dense areas, suggesting that N/G3BP1 structures are not sites of replication and transcription of the viral RNA (Fig. 5c). Lastly, we observed a colocalization of gRNA in some N/G3BP1 puncta (Fig. 5d), indicating that these structures are also sites of viral genome accumulation.

To better define these N/G3BP1 puncta, expansion microscopy was performed, providing at least a 4-fold linear increase in resolution. Figure 5e revealed the presence of large doughnut-shaped N structures with a hollow center reminiscent of N structures recently described by ref. 68. G3BP1 accumulated in these N ring structures, together with S. Colocalization with a marker of ERGIC (ERGIC3) further revealed that they are not structures devoid of membrane such as SGs, but are likely vesicles derived from the ER-Golgi intermediate. Moreover, proteins seemed non-uniformly distributed within these vesicles, possibly implying the presence of substructures. As observed earlier, N vesicles were found in the vicinity of dsRNA, but rarely co-localize with them (Fig. 5f). Overall, our data show that G3BP1 colocalized with 3 structural proteins and with the gRNA in structures containing ERGIC derived membrane, which likely represent sites of virion assembly and/or virion accumulation.

## G3BP proteins favor the formation of sites of virion assembly and/or accumulation

Next, we compared the spatial distribution of N and S proteins in control and G3BP1/2 KD cells after a single round of infection (at 10 hpi). The global distribution of viral proteins as well as the percentage of cells expressing N and S were similar in both conditions (Fig. 6a, b and Supplementary Fig. 6a), confirming that G3BP1/2 did not significantly impact viral infection, from virion entry to the translation of N and S proteins. In more than 90% of infected cells, N accumulated in puncta and downregulation of G3BP1/2 had no impact on the percentage of cells harboring these N puncta (Fig. 6a, c), suggesting that G3BP1/2 were not required for their formation. However, a 1.8-fold decrease in the number of cells with N/S puncta was observed in G3BP1/2 KD conditions, as compared to the control cells (Fig. 6a, c). Incidentally, this decrease correlated with the decrease of virions

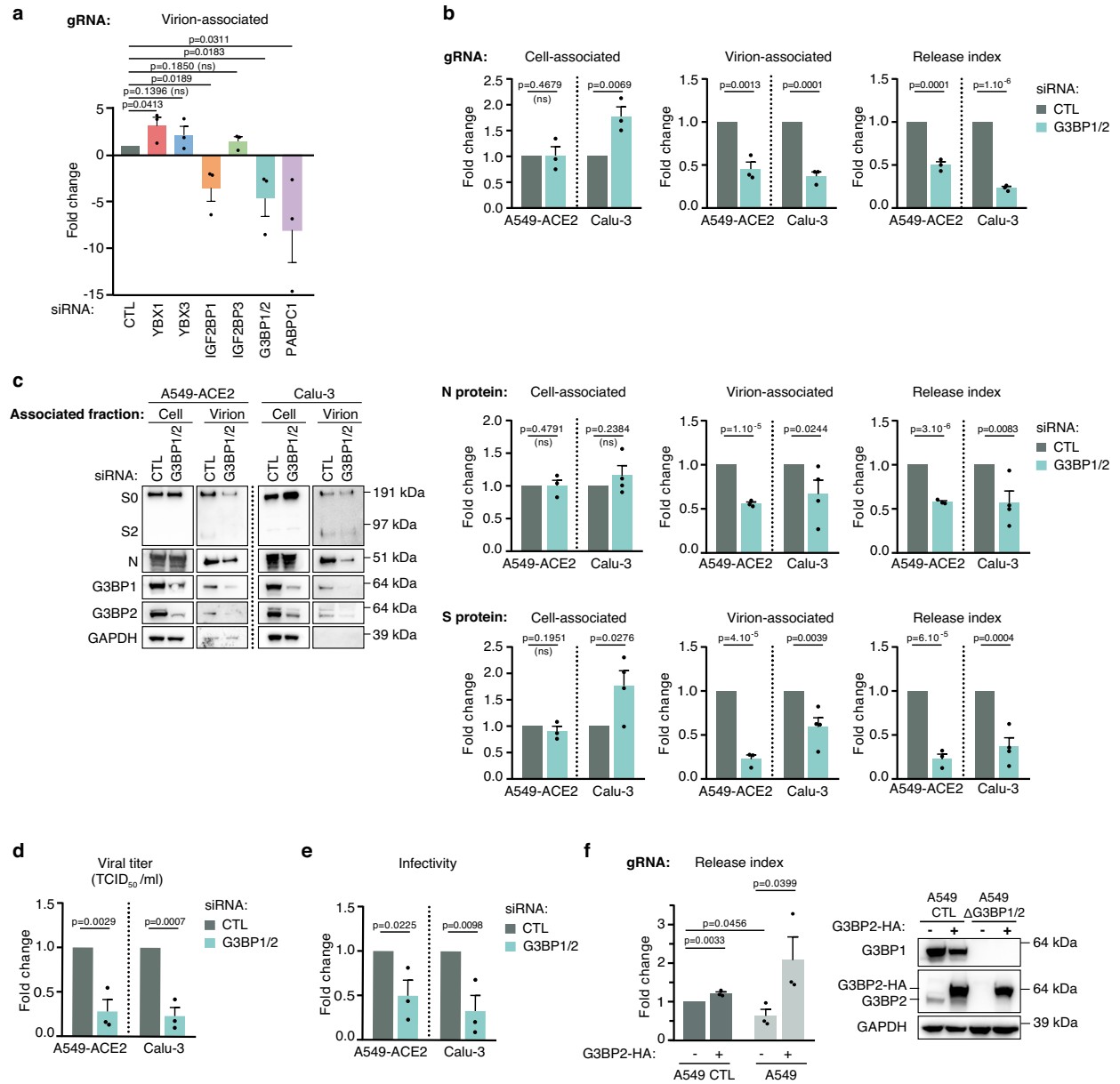

**Fig. 4 | G3BP proteins favor SARS-CoV-2 replication at a post-transcriptional level. a** A549-ACE2 cells transfected with siRNA control (CTL) or targeting indicated host proteins were infected with SARS-CoV-2 (MOI 0.01, 48 h). Viral genomic RNA (gRNA) level was quantified by qPCR in the virion-associated fraction. Data normalized to siRNA CTL are shown (*n* = 3 independent experiments). The KD efficiencies are presented in Supplementary Fig. 4b. **b–e** A549-ACE2 or Calu-3 cells transfected with siRNA CTL or targeting both G3BP1 and G3BP2 (G3BP1/2) were infected with SARS-CoV-2 (MOI 1, 10 h). Identical volumes of cell- and virion-associated fractions were used between CTL and G3BP1/2 KD conditions in all analyses. Data are normalized to siRNA CTL. **b** Quantification of cell- and virion-associated gRNA levels by qPCR. Cell-associated level (normalized to Actin RNA level), unnormalized virion-associated level and release index (calculated as the ratio of unnormalized virion-associated over unnormalized cell-associated level) of gRNA, are shown (*n* = 3 independent experiments). **c** N and S proteins were detected in both cell- and virion-associated fractions by immunoblotting with the indicated antibodies. For analysis, 1.25% and 25% of the cell- and virion-associated fractions, respectively, were loaded. A representative immunoblot is shown. Quantification of cell-associated levels relative to intracellular GAPDH, unnormalized virion-associated level and release index (calculated as the ratio of unnormalized virion-associated over unnormalized cell-associated level) of N and S are shown (*n* = 3 and *n* = 4 independent experiments for A549-ACE2 and Calu-3 cells, respectively). **d** Viral titers of virion-associated fractions were quantified by Tissue Culture Infectious Dose (TCID₅₀) assays (*n* = 3 independent experiments). **e** Infectivity of virion-associated fraction, defined as the TCID₅₀ normalized to the level of virions assessed by N ELISA quantification was calculated (*n* = 3 independent experiments). **f** Control (CTL) and double KO (ΔG3BP1/2) A549 cells, stably expressing (+), or not (−), G3BP2-HA were infected with SARS-CoV-2 (MOI 1, 10 h). Cell- and virion-associated gRNA was quantified by qPCR and release index, calculated as in **b** is shown (*n* = 3 independent experiments). Overexpression of G3BP2-HA and KO of G3BP1/2 were validated by immunoblotting. Data are presented as mean values +/− SEM; One-tailed unpaired Student's *t*-test was applied. Source data are provided as a Source Data file.

produced from G3BP1/2 KD cells (Fig. 4b, c). Overall, these data indicate that N can accumulate in puncta independently of G3BP1/2, but that G3BP proteins accumulate at and favor the assembly of N/S cytoplasmic structures.

To deepen these observations, we followed the distribution of N and S proteins in the cytosolic and membrane fractions of infected cells KD or not for G3BP1/2 at 24 hpi, after several rounds of infection. GAPDH and transferrin receptor (TF-R) were used as controls for the

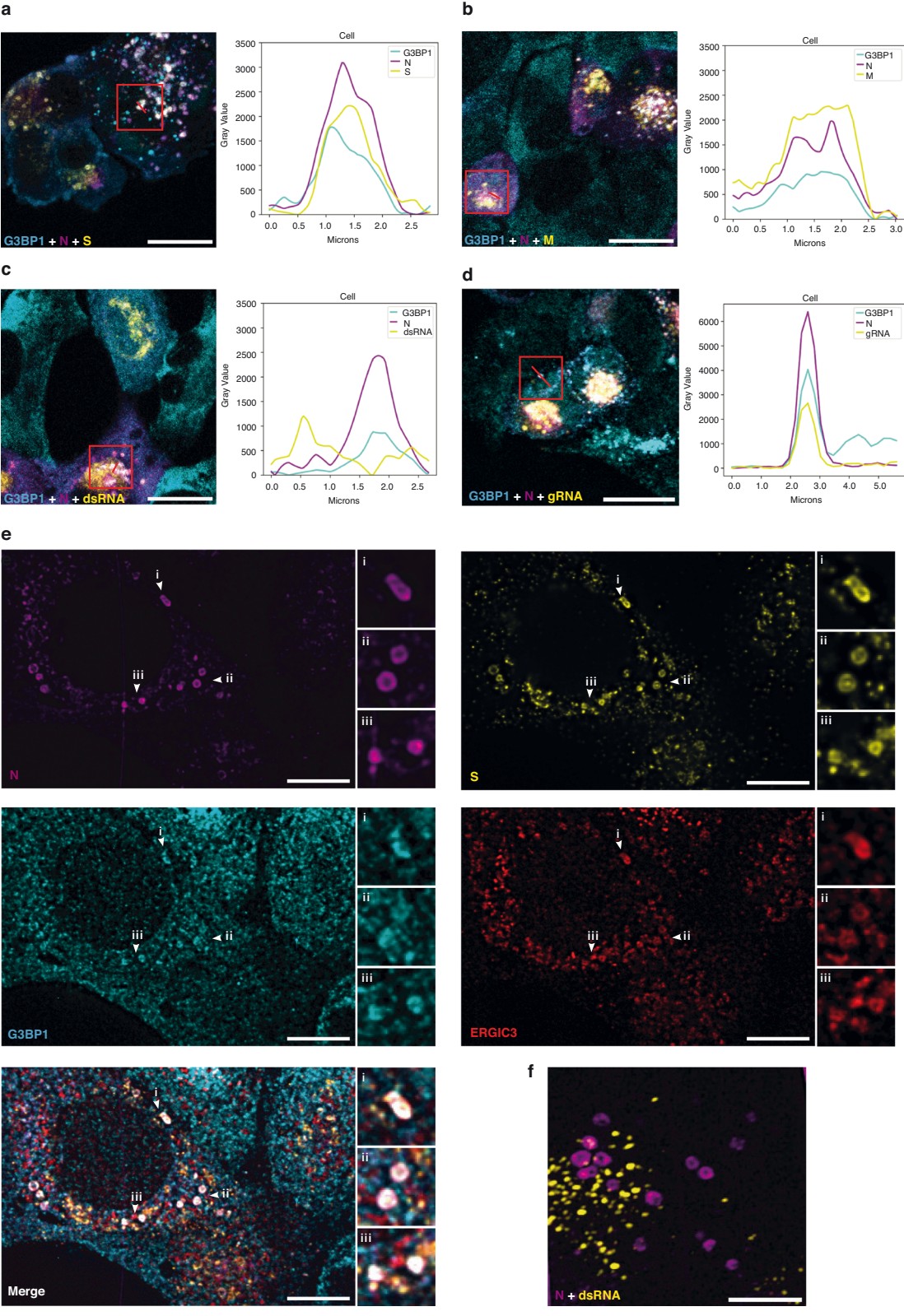

**Fig. 5 | G3BP proteins accumulate in N vesicles together with S and M proteins and viral genomic RNA.** A549-ACE2 cells were infected with SARS-CoV-2 (MOI 0.1) for 24 h. Representative images of the distributions of G3BP1 and of the viral proteins in cells harboring, or not, N puncta are presented in Supplementary Fig. 5a. **a**–**d** Details of the cellular distributions and colocalizations of G3BP1 and virion components in N puncta. Infected cells were stained with G3BP1, N, S, M and gRNA as indicated in the images. Images are representative of n = 3 (**a**–**c**) and n = 2 (**d**) independent experiments. **e** Expansion microscopy can resolve the N structures revealing their colocalization with G3BP1, S and ERGIC3 (images are representative of n = 3 independent experiments). Insets highlight colocalized structures. **f** N structures associate with, but do not colocalize with dsRNA (image representative of n = 2 independent experiments). All-scale bars indicate pre-expansion scales, 15 μm. Line scan of individual proteins and gRNA labeling in N puncta are shown. Source data are provided as a Source Data file.

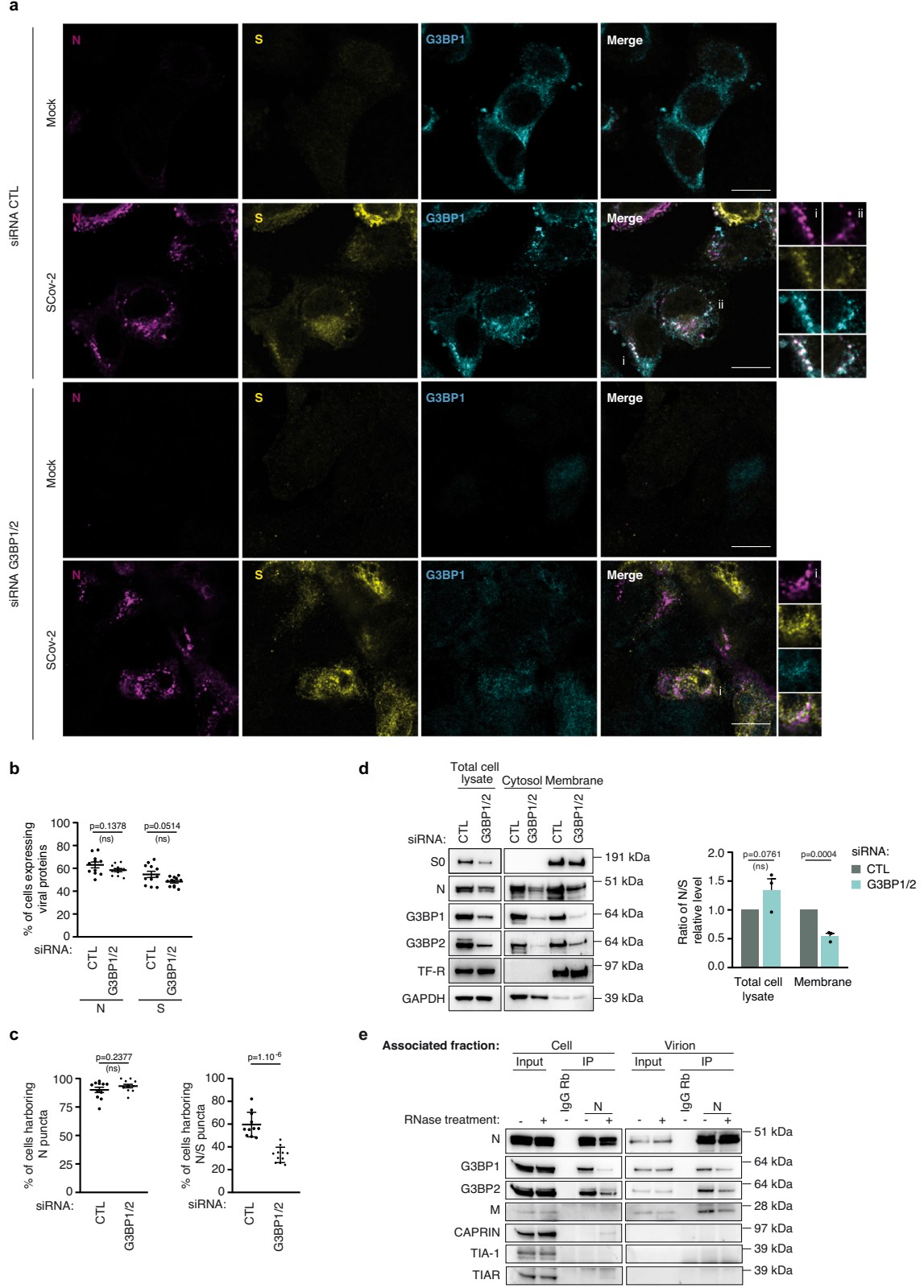

cytosolic and membrane fraction, respectively. As expected after several rounds of infection, depletion of G3BP1/2 dampened SARS-CoV-2 replication, leading to the decrease of N and S levels in total cell lysate as compared to control (Fig. 6d, Total cell lysate). The transmembrane protein S was exclusively detected in the membrane fraction (Fig. 6d). The presence of N in the cytosolic and membrane fraction is in line with its distribution in the cytosol and its

accumulation in membrane structures. Similarly, G3BP1 and G3BP2 were observed in the cytosolic and membrane fractions (Fig. 6d). Comparison of N and S revealed that for similar levels of S in the membrane fractions, a relative decrease of N was observed in G3BP1/2 KD cells compared to control cells (Fig. 6d). Western-blot quantification further confirmed that the N/S ratio was decreased by 1.9-fold in the membrane fraction in G3BP1/2 KD cells compared to the control,

**Fig. 6 | G3BP proteins favor the formation of vesicles where viral structural protein S accumulates. a–c** A549-ACE2 cells transfected with siRNA control (CTL) or siRNA targeting G3BP1 and G3BP2 (G3BP1/2) were either mock- or SARS-CoV-2-infected (MOI 1) for 10 h. **a** Cells were fixed and stained for N, S and G3BP1. A representative image of each condition is shown (scale bar = 15 μm). **b** The number of cells expressing N or S proteins expressed as a percentage of the total number of cells are shown. **c** Percentage of cells presenting N puncta among N expressing cells (left). Percentage of cells showing colocalization of S in puncta of N among all cells showing N puncta (right). Data in **b** and **c** are presented as mean values +/− SEM ($n$ = 1170 cells (si CTL) and $n$ = 1377 cells (si G3BP1/2) examined over 2 independent experiments). A two-tailed unpaired Student's $t$-test was applied. **d** A549-ACE2 cells were infected for 24 h. Total cell lysate, as well as cytosolic and membrane fractions were analyzed by immunoblotting with the indicated antibodies. A representative image of 3 independent experiments is shown (left). Relative N and S protein levels were estimated by quantifying bands using ImageJ. The ratio of N to S in total cell lysate and in the membrane fraction is shown (right). Data are presented as mean values +/− SEM; $n$ = 3 independent experiments. A one-tailed Student's $t$-test was applied. **e** Co-Immunopurification (IP) experiments were performed using anti-capsid N antibody or Rabbit IgG as a control from either cell- or virion-associated fraction in the absence (−) or presence (+) of RNAse treatment. The efficiency of RNAse treatment was verified in parallel on gRNA level by RT-qPCR. Proteins co-immunoprecipitated with N were detected by immunoblotting with the indicated antibodies. A representative image of 3 independent experiments is shown. Source data are provided as a Source Data file.

whereas this ratio was not significantly impacted in the total cell lysate (Fig. 6d). This suggests that G3BP1/2 favor the recruitment of N to S-containing membranes. G3BP1/2 physically interact with N[61,63,67,69,70]. In line with this, G3BP1 and G3BP2 co-immunoprecipitated with N in infected cells (Fig. 6e). In addition, we revealed that G3BP1/2 also interacted with N within virions, suggesting that this physical interaction is maintained throughout the assembly process (Fig. 6e). While this interaction was mostly mediated through RNAs in cells, it was resistant to RNAse treatment within lysed virions (Fig. 6e). One possible explanation is that the subset of G3BP1/2 that is incorporated inside virions directly interact with N. Alternatively, the cellular interactions of G3BP1/2 with N and the gRNAs may be reorganized during virion assembly. Of note, although we could not observe co-IP of M with N from cell extracts, as expected, a direct interaction was observed within virions (Fig. 6e). Altogether, we show that G3BP1/2 interact with N and/or viral RNP throughout the assembly process and participate in the recruitment of N to S-containing membrane vesicles, reinforcing the idea that G3BP1/2 participate in intracellular virion assembly.

## Discussion

To identify cellular factors involved in late stages of SARS-CoV-2 replication, we characterized host proteins associated with SARS-CoV-2 virions from A549-ACE2 and Calu-3 cells. In total, 356 host factors were found enriched in SARS-CoV-2 virions and were thus subjected to the following criteria: (1) involvement in molecular and cellular pathways linked to viral infections, (2) interactions with other virion-associated factors, (3) interaction with viral structural proteins and/or SARS-CoV-2 RNA and (4) presence in other viral particles. Distinguishing between proteins passively incorporated during virion assembly and actively recruited with functional relevance can be challenging. As described by ref. 71, we noticed a correlation between the LC-MS signal intensity of factors associated with virions and their protein abundances in A549 cells (R = 0.42; Supplementary Fig. 7a). However, when considering the degree of enrichment of factors in virion-associated fraction as compared to control fraction from mock-infected cells, we observed a significant reduction in the correlation with protein abundance (R = 0.13), in particular for SARS-CoV-2 RBP (R = −0.0093) (Supplementary Fig. 7b), suggesting that the degree of enrichment is a better indication of an active association with virions[71]. Isolating virions from other extracellular vesicles is also challenging. Consequently, virions isolated on sucrose cushion were further purified based on their affinity for ACE2 receptor. Overall, a total of 92 host factors were identified in virions produced from A549-ACE2 cells and subsequently confirmed in virions from Calu-3 cells and/or through ACE2-affinity capture. This included a subset of proteins that belong to phase separation-induced SG.

Importantly, G3BP1 and G3BP2, the major nucleators of SG, were specifically enriched in SARS-CoV-2 virions. G3BP1/2 interacted with N in both cellular and virion contexts, and subtilisin treatment of virions confirmed their incorporation within viral particles.

By sequestering viral components and engaging with innate immune responses, SG are thought to have antiviral properties[60,72]. To counteract SG assembly, viruses have thus evolved different strategies[58,59]. The interaction of SARS-CoV-2 N and nsp1 with G3BP1 was reported to reduce SGs assembly, alter G3BP1 mRNA-binding profile, modify its interactome and dampen host immune responses[12,61–63,65,67,70,73,74].

Herein, we showed that G3BP1 and G3BP2 downregulation decreased SARS-CoV-2 replication, indicating a proviral role for these two factors. Furthermore, knock-down of other SG factors associated with virions differently impacted SARS-CoV-2 replication, suggesting that the function of G3BP1/2 in viral infection may not be directly or solely related to their role in SG assembly (Fig. 4a). Our results differ from two recent studies that addressed the impact of G3BP1 KD on SARS-CoV-2 replication after multiple rounds of infection[61,62], but are consistent with three studies indicating that disruption of N and G3BP1 interactions decreases SARS-CoV-2 replication[63,64,74]. It is noteworthy that G3BP1/2 are multifunctional as in addition to triggering SG assembly, they can also enhance innate immune signaling, as recently reported for the RIG-I-mediated IFN response following SARS-CoV-2 infection[61]. Proviral functions of G3BP1/2 have also been described for several other RNA viruses such as chikungunya virus or dengue virus at various stages of viral replication connected to RNA metabolism[58,75]. Moreover, G3BP proteins may act as shock absorbers preventing excessive activation signaling and cell death[76]. It is therefore conceivable that various aspects of G3BP1/2 function were observed depending on the experimental conditions, including the cellular model, MOI and experimental timing. To gain a clearer understanding of the impact of G3BP1/2 on SARS-CoV-2 virion production, we exploited two pulmonary cell models, A549-ACE2 and Calu-3 cells at 10 hpi, e.g., before the formation of viral-induced SGs[65], or the initiation of the interferon response[21]. G3BP1 and G3BP2 are paralogous proteins that are structurally and functionally related, both triggering phase separation and SG assembly[75]. In the context of SARS-CoV-2 infection, we found both proteins interacted with N in infected cells and are incorporated within virions, likely reflecting their redundant/overlapping roles (Figs. 3e–g and 6e). Therefore, in contrast to other studies, we simultaneously KD the expression of both G3BP1 and G3BP2 either by siRNA or by CRISPR-Cas9 to avoid compensatory effects of single knockdown. Indeed, KD of one protein led to overexpression of the other, confirming the utility of simultaneously downregulating both proteins (Supplementary Fig. 4a)[40,41,66].

Our examination of the role of G3BP1/2 in SARS-CoV-2 at early times of infection (10 hpi) did not reveal any consistent impact on the intracellular level of gRNA, nor on the level of S and N proteins (Fig. 4b, c and Supplementary Fig. 4f, g). This is consistent with G3BP1/2 not being involved in early stages of SARS-CoV-2 replication, up to the translation of viral proteins. However and importantly, G3BP1/2 KD led to decreased production of infectious virions, reflected by a decrease in extracellular N, S and gRNA, viral titer and virion infectivity. This argues that G3BP1/2 participate in later stages of viral replication, such as gRNA packaging, virions assembly, or the egress of newly infectious

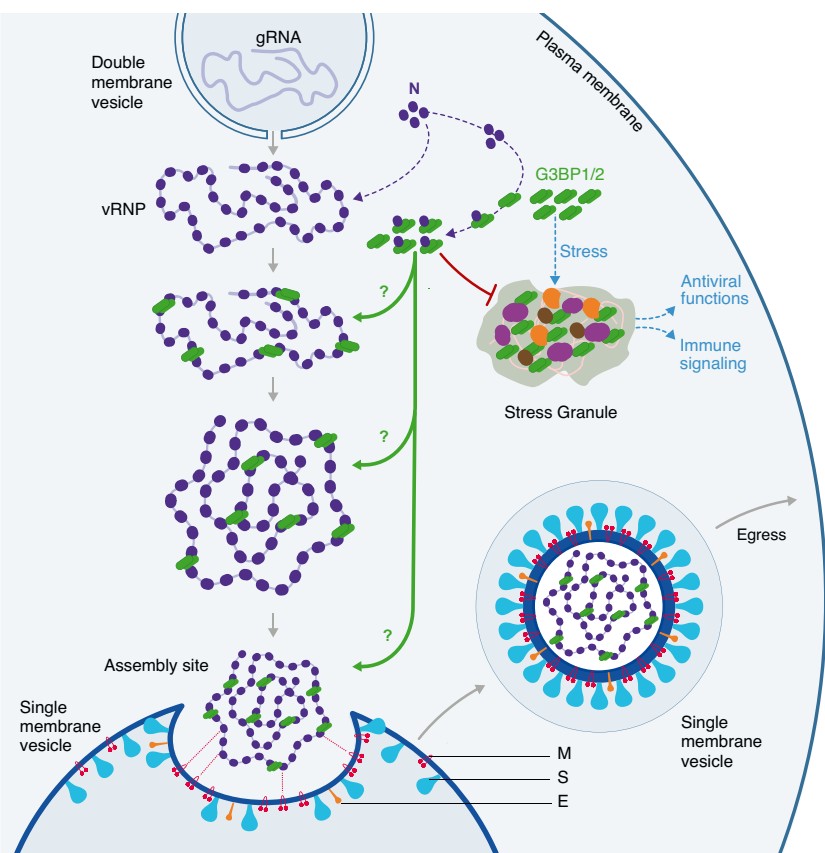

**Fig. 7 | Schematic model of the possible roles of G3BP proteins in the assembly of SARS-CoV-2 virions.** Upon stress and in the absence of SARS-CoV-2 infection, G3BP proteins can trigger phase separation of ribonucleoprotein complexes (RNPs) and induce stress granule (SG) assembly[40,55]. SG are sites of RNA storage. Upon viral infections, they can also sequester viral factors, control viral translation, stimulate immune signaling or prevent apoptosis;[58–60,72] Upon SARS-CoV-2 infection, N interacts with G3BP1/2 and viral genomic RNA (gRNA), preventing SG formation[12,61,62,65,67,70,73]. Our result indicate that G3BP1/2 also favor the assembly of new infectious virions possibly by: 1-participating to the viral RNP (vRNP) stabilization and/or reorganization, 2-inducing phase separation and vRNP compaction or 3-favoring the recruitment of the vRNP to the assembly site and its packaging.

viral particles (Fig. 4b–e and Supplementary Fig. 4d–h). Furthermore, this notion is reinforced by exogenous expression of G3BP2-HA fully restoring the release of SARS-CoV-2 virions in G3BP1/2 KO cells (Fig. 4f).

Our observation that N, S and M accumulated together in foci with gRNA and the ERGIC3 marker, strongly suggests that some of these vesicular structures are sites of virion assembly and/or virion accumulation (Fig. 5). As virion assembly occurs at ERGIC derived SMV in the vicinity of DMV[3,4,15,77], this probably explains the proximity of N/G3BP1 vesicles with dsRNA staining (Fig. 5c, f). Finally, our data suggest that G3BP1/2 play an important role in the formation of these sites. Indeed, G3BP1/2 KD led to a decrease of cells harboring N/S vesicles (Fig. 6c), correlating with the decrease of N found in the membrane fraction (Fig. 6d) and the decrease of virions produced in the supernatant of G3BP1/2 KD cells (Fig. 4b–e). Altogether, our data support a model in which G3BP1/2 interact with N and/or gRNA and favor virion assembly and/or virion accumulation in cytoplasmic vesicles within the cells, and thus the production of infectious virions in the supernatant (Fig. 7).

Whether G3BP1/2 carry out this function by mediating phase separation condensates remains to be determined. It was suggested that the propensity of N to phase separate contributes to the organization of the viral RNP complex and thereby to gRNA packaging and virion assembly[8,10,11,13,74]. Association of G3BP proteins with RNA is the central node of SG assembly into condensed RNP networks[39,43,78]. Thus, one appealing hypothesis is that by binding to N and gRNA, G3BP1/2

favor viral RNP aggregation/organization and therefore, packaging and assembly (Fig. 7).

Overall, the apparent contradictory role of G3BP proteins as antiviral factors through SG assembly and as pro-viral factors in virion assembly may be reconciled in a more dynamic and refined model, revealing a dual functionality of N/G3BP interactions: on the one hand N sequesters G3BP proteins to prevent antiviral SG formation and to circumvent subsequent antiviral immune responses and on the other hand, the virus hijacks the function of G3BP1/2 to favor production of infectious viral particle (Fig. 7). The development of drugs preventing N/G3BP interaction, such as a recently described peptide[63], may be particularly interesting to dampen viral replication, not only by favoring SG formation, but also by preventing SARS-CoV-2 virion assembly.

## Methods
### Cell culture
Vero-E6 cells (African green monkey kidney cells; ATCC CRL-1586) and A549-ACE2 cells (human tumorigenic lung epithelial cells[32]) were maintained in Dulbecco's Modified Eagle Medium High Glucose Glutamax (DMEM; Thermo Fisher Scientific) supplemented with 5% fetal bovine serum (FBS; Thermo Fisher Scientific), 1% penicillin/streptomycin (P/S; Thermo Fisher Scientific) at 37 °C with 5% CO2. Calu-3 (Human lung adenocarcinoma; ATCC HTB-55) were cultured in DMEM High Glucose Glutamax supplemented with 10% FBS, 1% P/S and 1% non-essential amino acids (NEAA; Thermo Fisher Scientific) at 37 °C

with 5% CO2. The parental A549 and double knock-out (ΔG3BP, clone 5) cells were kindly provided by ref. 76. For the generation of cells stably expressing G3BP2-HA, G3BP2-HA insert was subcloned from pLVX-G3BP2-HA-EF2a-IRES-ZsGreen, generously provided by Dr. Ali Amara, into pLVX-EF1a-IRES-Puro (Clontech) using the NotI and BamHI restriction sites. HEK293T (human embryonic kidney cells, ATCC CRL-3216) were transfected with 2.25 µg of psPAX2 (Addgene # 12260), 0.75 µg of VSVg (Addgene # 14188) and 3 µg of pLVX-G3BP2-HA-EF1a-IRES-Puro or pLVX -EF1a-IRES-Puro empty vector as control using polyethylenimine (PEI, Polysciences). Supernatants were collected 48 h post-transfection and filtered with filtropur S 0,45 µm (Sarstedt). Parental A549 and A549ΔG3BP cells were transduced with supernatants for 24 h before selection in 1,5 µg/ml of puromycin (Thermo Fisher Scientific) for 5 days.

## Viral production

SARS-CoV-2 strain 220_95 (EPI_ISL_469284) was isolated from nasopharyngeal swab specimens collected at Service de Virologie (Hospital Saint Louis, Paris)[79] and propagated in Vero-E6 cells in DMEM High Glucose Glutamax supplemented with 2% FBS and 1% P/S for 72 h and was concentrated through a TNE 20% sucrose cushion (Tris pH 7.4 50 mM, NaCl 100 mM, EDTA 0.5 mM, sucrose 20%) by ultracentrifugation at 45,000 g for 2 h at 4 °C. Pellets were resuspended in HNE (Hepes 5 mM, NaCl 150 mM, EDTA 0.1 mM) buffer. Virus titer was determined by plaque assays in Vero-E6 cells and expressed as PFU per ml.

To produce virions from A549-ACE2 and Calu-3 cells, cells were infected at MOI 0.01 for 48 h and 40 h, respectively. Supernatants were collected and virions were isolated through a TNE 20% sucrose cushion by ultracentrifugation at 45,000 g for 2 h at 4 °C. Pellets were resuspended in Triethylammonium Bicarbonate Buffer (TEAB; Merck) and filtered through Amicon Ultra15 100 kDa (Merck) by centrifugation at 3000 g for 20 min at 4 °C.

Stripping of virions with subtilisin was performed as described in ref. 57. Briefly, virions isolated from A549-ACE2 cells through 20% sucrose cushion were resuspended in HNE buffer supplemented, or not, with 1 mg/mL subtilisin A (Merck) for 15 min at 37 °C. Subtilisin treatment was stopped by addition of DMEM containing 10% FBS, 20 mM EDTA (ThermoFisher Scientific), 5 mM Phenylmethylsulfonyl fluoride (PMSF). Stripped virions were further isolated on a 20% sucrose cushion followed by ultrafiltration as previously described.

To capture virions by affinity with biotinylated ACE2 receptors, virions isolated on sucrose cushion were further pre-cleared overnight with 1 mg of Dynabeads MyOne streptavidin T1 (Thermo Fisher Scientific). Virions were then incubated 2 h at 4 °C with 20 µg of Biotinylated Human ACE2/ACEH Protein, His, Avitag™ (ACROBiosystems) or 3000 pmol of D-Biotin (Thermo Fisher Scientific) as a control. After addition of 5 mg of beads to each condition, viruses were placed 15 min at 4 °C under rotation and then washed 3 times with HNE, 250 mM sucrose. ACE2-biotin or control biotin captured virions were eluted from streptavidin-beads in HNE, 250 mM sucrose and 0.1% SDS (Thermo Fisher Scientific) at 95 °C for 5 min.

## Liquid chromatography-coupled Mass spectrometry analysis

Isolated virions were boiled in resuspension buffer supplemented with SDS (2% final concentration) at 95 °C for 5 min. Bottom-up experiments tryptic peptides were obtained by S-trap Micro Spin Column according to the manufacturer's protocol (Protifi). Briefly: proteins of the above eluate were diluted 1:1 with 2x reducing-alkylating buffer (20 mM TCEP, 100 mM Chloroacetamide in 400 mM TEAB pH 8.5 and 4% SDS) and left 5 min at 95 °C to allow reduction and alkylation in one step. S-trap binding buffer was applied to precipitate proteins on quartz and proteolysis took place during 14 h at 37 °C with 1 µg Trypsin sequencing grade (Promega). After speed-vacuum drying of eluted peptides, these were solubilized in 0.1% trifluoroacetic acid (TFA) in 10% Acetonitrile (ACN).

LC-MS analyses were performed on a Dionex U3000 HPLC nanoflow chromatographic system (Thermo Fisher Scientific) coupled to a TIMS-TOF Pro mass spectrometer (Bruker Daltonik GmbH). One µL was loaded, concentrated, and washed for 3 min on a $C_{18}$ reverse phase precolumn (3 µm particle size, 100 Å pore size, 75 µm inner diameter, 2 cm length, Thermo Fisher Scientific). Peptides were separated on an Aurora C18 reverse phase resin (1.6 µm particle size, 100 Å pore size, 75 µm inner diameter, 25 cm length mounted onto the Captive nanoSpray Ionisation module, (IonOpticks)) with a 60 min overall run-time gradient ranging from 99% of solvent A containing 0.1% formic acid in milliQ-grade $H_2O$ to 40% of solvent B containing 80% acetonitrile, 0.085% formic acid in $mQH_2O$. The mass spectrometer acquired data throughout the elution process and operated in Data Dependent Analysis with Parallel Accumulation and Serial Fragmentation (DDA PASEF) mode with a 1.9 s/cycle, with Timed Ion Mobility Spectrometry (TIMS) enabled and a data-dependent scheme with full MS scans in PASEF. This enabled a recurrent loop analysis of a maximum of up to 120 most intense nLC-eluting peptides which were Collision Induced Dissociation-fragmented between each full scan every 1.9 s. Ion accumulation and ramp time in the dual TIMS analyzer were set to 166 ms each and the ion mobility range was set from $1/K0 = 0.6$ Vs $cm^{-2}$ to 1.6 Vs $cm^{-2}$. Precursor ions for MS/MS analysis were isolated in positive polarity with PASEF in the 100–1700 m/z range by synchronizing quadrupole switching events with the precursor elution profile from the TIMS device. The cycle duty time was set to 100%, accommodating as many MSMS in the PASEF frame as possible. Singly charged precursor ions were excluded from the TIMS stage by tuning the TIMS using the otof control software, (Bruker Daltonik GmbH). Precursors for MS/MS were picked from an intensity threshold of 1000 arbitrary units (a.u.) and re-fragmented and summed until reaching a 'target value' of 20,000 a.u., while allowing a dynamic exclusion of 0.40 min elution gap.

The mass spectrometry data were analyzed using Maxquant version 1.6.17 for Label-Free Quantification (LFQ) analysis[80]. The database used was a concatenation of *Homo sapiens* sequences from the Swissprot database (release june 2020: 563,972 sequences; 203,185,243 residues), and the 16 proteins of the SARS-CoV-2 virus and an in-house list of frequently found contaminant protein sequences. The enzyme specificity was trypsin's. The precursor and fragment mass tolerances were set to 20 ppm. Oxidation of methionines was set as variable modifications while carbamidomethylation of cysteines was considered complete. Second peptide search was allowed and minimal length of peptides was set at 7 amino acids. False discovery rate (FDR) was kept below 1% on both peptides and proteins. Label-Free protein Quantification (LFQ) was done using both unique and razor peptides. At least 2 such peptides were required for LFQ. For differential analysis, LFQ results from MaxQuant were quality-checked using PTXQC[81], then imported into the Perseus software (version 1.6.15)[80] for statistical analysis.

## Proteomic statistics

Proteins annotated as "Reverse" or "Only identified by site" were excluded from analysis as well as contaminant proteins according to MaxQuant and to the common Repository of Adventitious Proteins cRAP (https://www.thegpm.org/crap/). LFQ intensities of quantified proteins were then log2-transformed.

For A549 and Calu-3 cell supernatants as for capture experiment, only proteins with at least 3 LFQ valid values in one of the groups were considered. Missing values were then replaced by random numbers drawn from a normal distribution according to default parameters (width = 0.3 – down shift = 1.8). A one-tailed paired Student t-test associated with a Benjamini–Hochberg false discovery rate adjustment was performed in order to identify significant differential molecules

among the proteins upregulated in SARS-CoV-2 samples compared to uninfected ones.

## In silico functional analysis

Protein–protein interaction networks were built using STRING v.11.5[48] based on Experiments and Databases interaction sources with default setting parameters. Data were imported and visualized using Cytoscape (v.3.8.2)[82]. Gene Ontologies over-representation analyses were performed with DAVID online tool (updated version 2021, https://david.ncifcrf.gov/)[83,84]. GOTERM_BP_5, GOTERM_MF_ALL functional annotation clustering and KEGG pathway annotation charts were retained. The top 10 results with an adjusted *p*-value (Benjamini–Hochberg) <0.05 were plotted with RStudio (version 2023.06.1) using ggplot2 package (version 3.3.5). Factors were considered as known RBP based on the RBPbase (https://rbpbase.shiny.embl.de/).

## Quantitative RT-PCR

For quantification of virion-associated RNA, total RNA of infected cells was extracted from half of the cell-associated lysate with RNeasy mini kit (Qiagen), treated with TURBO DNase (Thermo Fisher Scientific) and retro-transcribed using the High Capacity cDNA Reverse Transcription kit (Thermo Fisher Scientific) according to the manufacturer instructions. cDNAs were quantified with the SsoAdvanced Universal SYBR Green Supermix (Bio-Rad) using specific sets of primers (Supplementary Table 1). For quantification of virion-associated RNA, 5% of the collected supernatants were lyzed in PBS, 1% Triton X-100 (Merck) during 30 min. RNA was retro-transcribed and cDNA was quantified with Luna Universal One-Step RTq-PCR (New England BioLabs) using primers specific of SARS-CoV-2 gRNA (Supplementary Table 1).

## Co-Immunoprecipitation

A549-ACE2 cells were infected at MOI 0.01 for 48 h. Cells and viral supernatants were lysed in 50 mM Tris pH 8, 150 mM NaCl, 0.25% Triton X-100 and complete protease inhibitor cocktail (Merck). Cells lysates were cleared by centrifugation for 20 min at 15,000 g at 4 °C. Cells and viral lysates were treated or not with RNase A/T1 Mix (Thermo Fisher Scientific) for 15 min at 37 °C. Samples were incubated with 1 µg of nucleocapsid antibody or 1 µg of rabbit IgG control antibody and placed overnight at 4 °C under rotation. After addition of 1.5 mg of Dynabeads protein G (Thermo Fisher Scientific) for 5 hours, samples were washed 3 times with 50 mM Tris pH 8, 150 mM NaCl. Beads were resuspended in Laemmli 1X (Thermo Fisher Scientific) and boiled at 70 °C for 10 min.

## Immunoblotting

Cells were lysed with RIPA buffer (50 mM Tris, pH 8, 150 mM NaCl, 1 mM EDTA, 1% Triton X-100, 1% DOC and 1% SDS) containing complete protease inhibitor cocktail and cleared by centrifugation for 20 min at 15,000 g at 4 °C. Proteins were resolved by SDS-polyacrylamide gels electrophoresis using Bolt 4 to 12%, Bis-Tris, 1.0 mm minigel (Thermo Fisher Scientific), transferred to hydrophobic polyvinylidene difluoride membranes 0.45 µm (PVDF, Merck) and detected by immunoblotting using appropriate antibodies (Supplementary Table 2). Protein bands were detected using either Amersham ECL Select Western blotting (Merck) or Immobilon Crescendo Western HRP Substrate (Merck) detection reagent and revealed with Fusion FX (Vilber). Signals were quantified using Fiji.

## siRNA and CRISPR-Cas9 treatments

A549-ACE2 and Calu-3 cells were transfected with ON-TARGETplus SMARTpool siRNA (Dharmacon) (Supplementary Table 3) at a final concentration of 30 nm using Lipofectamine RNAimax (Thermo Fisher Scientific), according to the reverse transfection procedure described in the manufacturer's recommendation. Cell viability of siRNA-transfected cells in both mock-infected and SARS-CoV-2-infected conditions was assessed using the Cell-Titer Glo Luminescent Cell Viability assay (Promega) following the manufacturer's instructions.

For lentiCRISPR production, 10 cm² dishes of 80% confluent HEK293T cells were transfected with 2.25 µg of psPAX2 (Addgene # 12260), 0.75 µg of VSVg (Addgene # 14188) and 3 µg of LentiCRISPRV2 plasmid (Supplementary Table 4) using PEI. LentiCRISPR were collected 48 h post-transfection and filtered with filtropur S 0.45 µm. A549-ACE2 cells were transduced with lentiCRISPR for 24 h before selection in 1.5 µg/ml of puromycin for 5 days.

## N protein ELISA quantification

Cell- and virion-associated N protein levels were quantified using COVID-19 / SARS-COV-2 Nucleocapsid Protein ELISA kit (RayBiotech), as described in the manufacturer's recommendation.

## TCID50/mL

Virus titers were quantified by challenging A549-ACE2 cells with serial dilutions of ultracentrifuged supernatant. The 50% Tissue Culture Infectious Dose (TCID50/mL) was calculated by examining cytopathic effects on the infected cells.

## Electron microscopy

Virions isolated on sucrose cushion were fixed with 2.5% of glutaraldehyde. Negative stain electron microscopy was performed by applying SARS-CoV-2 particles to a glow-discharged formvar-carbon coated EM grid (homemade) followed by staining with 1% uranyl acetate. Imaging was performed on a JEOL 1011 TEM with a Gatan Orius 1000 CCD Camera.

## Immunostaining

Cells were fixed with a 4% PFA solution in PBS for 10 min at room temperature (RT), washed with PBS and then permeabilized by applying 0.1% (v/v) Triton X-100 in PBS for 15 min at RT. After blocking in PBS-5% BSA, cells were stained with primary antibodies (Supplementary Table 2) in PBS-1% BSA for one hour at RT or overnight at 4 °C. After 3 washes, cells were incubated with a secondary antibody for one hour at RT, washed and stored in PBS.

## smFISH

Probes specific for SARS-CoV-2 genomic RNA were designed using the designer tool from BioSearch Technologies (https://www.biosearchtech.com) with a masking level of 5 and a minimum spacing length of 2 nt between each probe. A total of 30 probes of 18 nt in length were conjugated with TAMRA-C9 (Supplementary Table 5). Immunostaining was performed as described before. All buffers contained RNase inhibitor to prevent RNA degradation. Probes were diluted in a buffer composed by 0.1 g/ml of dextran sulphate, 2x SSC and 1% formamide and were let hybridize overnight at 37 °C to a final concentration 25 nM. The following day, cells were washed with 2x SSC, 10% formamide and imaged in PBS.

## Expansion of immunostained cells

Immunostained samples that were meant for expansion microscopy were incubated with 0.1 mg/ml Acryloyl-X SE (AcX) in 150 mM sodium bicarbonate (pH 8.3) overnight at RT. Cells were then washed twice, each time for 15 min, with PBS. The gelation solution was applied on the cells in a pre-built gelation chamber, and it was left polymerized in a humidified environment at 37 °C for 1 h. The hydrogel was then incubated overnight at RT in a digestion buffer containing Proteinase K (8 U/ml), and expanded three times in mQH$_2$O, each time for 20 min. After expansion, gels were mounted in Ibidi µ-slide 2 well glass bottom plates coated with 0.1% (w/v) poly-L-lysine and imaged in MilliQ water. See[85] for buffers composition.

## Image acquisition and analysis

Images of both fixed and expanded cells were acquired in an Andor spinning disk confocal equipped with an Andor iXon 897 EMCCD camera or with Microscope IXplore spinning disk Olympus, using a 60x, oil objective. For each xy location of fixed samples, a z-stack of 8 steps was taken, 1.36 μm each step. For expanded samples, the z-stack was either composed by 5 steps, 0.75 μm each, or 21 steps, 0.25 μm each. Fixed, non-expanded cells were used to quantify protein colocalization. For each position, single z-slices were chosen and from these a mask selecting only infected cells was obtained in Fiji. Pearson's coefficients of single cells were calculated from thresholded images using in-house Python programs. The intensity analysis of proteins at the level of nucleocapsid puncta was performed in Fiji on images of fixed, non-expanded cells. All data were plotted in Python.

## Subcellular fractionation

A549-ACE2 cells were transfected with siRNA and infected with SARS-CoV-2 at MOI 0.1. Cells were harvested 24 hpi. Two percent of the cells were lysed in RIPA buffer for immunoblot analysis. The rest was resuspended using ice-cold isotonic buffer (20 mM Hepes, pH 7.4, 250 mM sucrose, and 1 mM EDTA) supplemented with complete protease inhibitor cocktail and incubated for 15 min on ice. The resuspended pellet was then passed through a A dounce homogenizer 20 to 25 times for homogenization before clarification by centrifugation 5 min at 3000 g at 4 °C. Extracts were ultracentrifuged at 100,000 g for 1 h at 4 °C. The crude membrane fraction was resuspended in Laemmli buffer 1X for Western blotting analysis.

## Statistical analysis

The statistical details of all experiments are reported in figures legends, including statistical analysis, error bars, statistical significance, and exact n number. Statistics were performed using GraphPad Prism 6 software.

## Reporting summary

Further information on research design is available in the Nature Portfolio Reporting Summary linked to this article.

# Data availability

All relevant data supporting the key findings of this study and any associated accession codes and references are available within the article and in the Supplementary Information Files or from the corresponding authors upon request. Source data are provided with this paper. The mass spectrometry proteomics data generated in this study have been deposited in the ProteomeXchange Consortium via the PRIDE[86] partner repository under accession code PXD038321, PXD045406, PXD045409. Source data are provided with this paper.

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

## Acknowledgements

This work was supported by the Agence Nationale de la Recherche (ANR RA-Covid-19; ANR-20-COV1-000) to S.G.M. and L.M., the Agence Nationale de la Recherche sur le SIDA et les hépatites virales-Maladies Infectieuses Emergentes (ANRS-MIE) to S.G.M., the Christine Mohrmann Fellowship and the Netherlands Organization for Scientific Research (NWO) ENW-XS award (OCENW.XS21.2.050) to M.M.K.H. and by the DIM Thérapie Génique Paris Ile-de-France Région, IBiSA, and the Labex GR-Ex to Proteom'IC core facility. M.M.K.H. also acknowledges support from the French Ministry of Foreign Affairs, the French Embassy in the Netherlands and the French Ministry of Higher Education and Research through the Descartes Huygens Prize. We thank Virginie Salnot and Thomas Guilbert for their technical support and Monsef Benkirane, Ali Amara, Stéphane Emiliani, Mélissa Ait Said, Katy Janvier, Gordon Langsley, Phillipe Roingeard, Emilie-Fleur Gautier and Ismael Boussaid for helpful discussions.

## Author contributions

Conceptualization: S.G.M., L.M., and C.B.T., Funding acquisition: S.G.M., L.M., and M.M.K.H. Experimental design: S.G.M., M.M.K.H., D.J., V.P., F.G., and L.S. Investigation (did the experiments): E.M., L.S., C.G., J.B., A.S., C.A., S.G.M., and C.B.T. Experimental data analysis: E.M., L.S., S.G.M., C.G., M.M.K.H., and D.J. Bioinformatic analysis: C.A. and M.L. Work supervision: S.G.M., M.M.K.H., L.M., and F.G. Writing, review and editing of the manuscript: E.M., L.S., C.A., C.G., D.J., J.B., M.L., F.G., A.S., V.P., L.M., C.B.T, M.M.K.H., and S.G.M.

## Competing interests

The authors declare no competing interests.
