## [Peer Review File · Nature Communications]

Proteomic analysis of SARS-CoV-2 particles unveils a key role of G3BP proteins in viral assemblyREVIEWER COMMENTS

Reviewer #1 (Remarks to the Author):

Murigneux et al present here an interesting story, in which they analyse the proteome of SARS-CoV-2 viral particles using two approaches. Their results reveal a few dozens of cellular proteins, most of them RNA-binding proteins (RBPs). Amongst them, authors noticed the presence of the stress granule forming proteins G3BP1 and G3BP2. KD of these proteins induce a decrease in the formation of the viral particles and they observe co-localisation with N in cells and a physical interaction that is RNase sensitive in cells and insensitive in viral particles. While interesting, I think there are points that require further work.

1. The limitations of virion proteomic analysis (see below) are not described and contextualized and I think it is important for the reader to understand the limitations of the datasets that they are looking at and help in the process of making educated guesses in the selection of candidates. I understand that the data looks good for the standards of viral particle analysis, but still, many of the discovered proteins will likely be bystanders.

2. Analysis of the proteome of the viral particles. I think overall, the authors have done a good job when using the sucrose gradient approach, but I don't understand some of the decisions taken. For example, authors applied a sound approach using well-established label free quantification and estimation of the false discovery rates (FDRs). However, they take the arbitrary decision of following a semi-quantitative approach in which only proteins identified in the viral particles are considered. I think this make little sense from the proteomic point of view. When considering a given protein in two samples, identification and quantification is a matter of signal and noise as well as peptide crowding at a given point of the chromatogram. Black and white situations are not necessarily the best, because they can represent cases in which one protein is detected at trace level in condition A and is absent in condition B. This would be considered as an optimal candidate although it is actually not a good one. Hence, the peptide intensity should be factored in some way. The quantitative approach does that (particularly if applying noise imputation and FDR correction) by generating ratios that are dependent on intensity levels in both samples. I understand authors are reluctant because the intrinsic properties and limitations of the analysis of viral particles, however, these limitations can be counteracted in other ways as described below. However, I don't think that the strategy of discarding the quantitative approach and replacing it by a semiquantitative approach improves the data in any way.

3. The problem of viral particle proteomics is that virions assemble inside the cell and in the process will inevitably take a portion of the cytosol (or compartment where this happens) that can perfectly contain protein with no role in infection. Indeed, comparison of the intensity of proteins in virions purified in previous studies correlate very well with that of the cell proteome. Therefore, there is different ways to benchmark the results that are not considered here, including whole proteome, generation of particles in other cell lines, etc. Authors try to overcome this by applying a second way of isolation of virions, using biotinylated ACE2. Unfortunately, this experiment seems tremendously inefficient and is not well controlled (for example capture using the biotinylated ACE with the preparations from uninfected cells). Therefore, the results aren't convincing and can account only for a minority of the proteins identified in the virions. The experiment is not particularly useful and raises questions about isolation efficiency and missing controls to account for biases and I would personally not trust the data provided.

Another limitation is that proteins can be soluble or trapped at the surface of the virion. Authors tried to minimize this by using an amicon purification step, that I am sure helped but will not eliminate proteins forming large (>100KDa) complexes or attached to the surface of the virion. Tryptic digestion of the sample before the gradient separation would help as the envelope is supposed to protect intravirion proteins. This latter point might be difficult to counteract at this point, but should be at least used to determine if the proteins characterized in Figure 4a are inside the viral particle.

4. Authors detect initiation factors and ribosomes inside the viral particles. This can be a consequence of defective exit of translation before packaging or can be a mechanism to 'label' the viral RNA for translation in the newly infected cell. Analysis of the coverage (which ribosomal and eIF proteins are detected) and intensities of these proteins would shed light on this idea. Are they bystanders or functionally necessary (I understand it is not the focus of the paper, so no experimental validation is required).

5. There is a discussion about whether G3BP1/2 would play its role in infection by producing phase transition or not. This however isn't clear as G3BP1/2 are active RBPs with role in infection in their 'soluble' form. The fact that other SG proteins are not in the virions suggest that the role of

G3BP1/2 are likely to be in their soluble form. Analysis of TIA1, TIAR, etc in the virions and IPs (Figure 6 – intracellular vs virions) would help to determine whether is a lack of incorporation or just that the sensitivity is not enough to detect them. Also the fold change of these proteins in the proteomic analysis should be provided as they might be enriched in a not significant manner or disqualify for the on/off criteria that the authors applied.

6. Phenotypes of G3BP1/2, IGF2BP1, PABPC1. These are all very central to RNA metabolism. Therefore, it would not be surprising that their effects in infection are actually indirect (as each regulates hundreds/thousands of cellular transcripts). I think these experiments should be better characterized. For example, do the overexpression of these proteins affect infection? Can the phenotype be rescued? What is the effect of the KD in the cell gene expression (any available RNAseq in KD cells? eCLIP?)?

7. G3BPs are antiviral in several other studies. What are the experimental differences with the present study? How these results can be conciliated?

8. Figure 4. What about viral titre, plaque morphology, etc ? G3BPs should then reduce the viral titre, and may have additional effects in a second round of infection (reflected in plaque number and size).

9. Figure 5. What is the localisation of N and other viral proteins in G3BPs KD? Controls of the antibodies in the mock cell and KD cells should be included. The analysis of the microscopy data should be more systematic, providing Pearson correlation of the different channels and quantification of N in WT and KD.

10. Figure 6d shows that KD of G3BPs decreases N levels in both whole cell proteome, cytosol (particularly) and membranes. WB does not reflect the quantification showed in the right panel, in which cytosol quantification is not included. The results showed disagree with the ELISA results presented previously and suggests that KD affect N to some extent. Indeed a slight decrease of N is also observed in inputs of panel e.

11. Figure 6e. The observed differences may well have a different explanation, which is compaction. Viral RNA might be more accessible to RNases when is not or partially associated to N than when it is fully compacted and recovered into viral particles. Analysis of the RNA after RNase in both conditions should be provided.

MINOR POINTS

1. Abstract: the two strategies are not described. That part does not read well.

2. Page 2: ongoing pandemic > this might not age well (hopefully). I recommend removing the 'ongoing'.

3. The downregulation of G3BP1 is not evident in the KD of G3BP2

4. Figure 6A. Improve labelling. It is not evident how upper an bottom panel differ without reading the legend.

Reviewer #2 (Remarks to the Author):

Overview:

Murigneux et al. have performed a very interesting and informative study, that includes appropriate controls, sound methodology, and comprehensive analyses with orthogonal techniques that provide new knowledge in the field. This work identified 175 host proteins associated with SARS-2 virions using distinct methodologies, and focussed efforts on characterizing how two RNA binding proteins (G3BP1 and G3BP2), which normally are found in stress granules, can favor virus assembly and act in a proviral manner. This work is novel and has not been reported elsewhere, reveals new molecular mechanisms in virus assembly, and highlights the interaction of viral N proteins and G3BP proteins as a new potential antiviral target. Overall this manuscript was well-written (it was a pleasure to read!) and provides new knowledge to the field with broad applications that are also very timely in the current stage of the ongoing COVID-19 pandemic.

Comments to be addressed by the authors:

1. Line 194-195: I'm confused about the language here. Were these tetraspanins not detected in any of the analyses at all, or were they only absent from your SARS-CoV-2 network analyses? I would be surprised that exosomes were completely eliminated from your preparations, as there are exosomes that would be similar size to your virus particles that would co-purify with your ultracentrifugation and filtration protocols. Also, many exosome markers can also be present on virion surfaces, as they are not exclusive to only exosomes, particularly for viruses like SARS-2

that bud. Please clarify this in the writing.

2. Line 198 – please change ‘quasi totality’ to a more definitive/descriptive word choice.
3. Line 218-220: Please re-word for clarity. I had trouble understanding what was meant here.
4. Line 224-229 and Suppl Fig 1C: More details of this analysis would make the comparison of your data on SARS-2 to other viruses more useful. For example, which 21 enveloped viruses were studied here? What methods were used in the Dicker et al and Gale et al. papers? It’s surprising that common surface proteins are not shown in Fig 1C, particularly since these are enveloped viruses, including for example, tetraspanins. Can you explain this?
5. Fig 3e immunoblots: Please provide more details of how these expts were performed to control for loading and comparisons of protein across samples. How much protein was loaded in cell lysate lanes vs virion lanes? How was the virion input normalized? Similarly, how much total protein was loaded in Fig 3F, and how was this controlled/normalized for flow through vs affinity-captured virus? What was the total virus ‘input’ in lane 1? Please provide more of these details for the reader to appreciate how much protein (virus lysate) is being analyzed and detected.
6. Lines 301-302 & Fig 4A: why did KD of YBX1 and YBX3 result in increased viral RNA? Was this expected? Can you comment on what is different about these proteins vs the others that seem to have the opposite effect (proviral)?
7. Line 318: why do the authors choose to refer to the virus-associated N protein as ‘extracellular’ levels in the main text (as on line 318), but ‘virions’ in the figure panel (as in Fig 4D). I would think it should be harmonized, and more accurately described a cell-associated or virion-associated.
8. Fig 4C: Again, more details on how this assay was performed are needed to enable accurate interpretation of the results. What was the cell input and viral input (amount of protein)? how was input controlled and where/what is the loading control for virion lanes? What protein measurement in viruses was used to normalize the observed reduction of virus-associated N protein (and S protein) shown in the blots and plotted in the bar graphs for 4C?
9. The legend for Fig 7 should indicate what the DMV and SMV are for clarity to a non-specialist. Or if space permits, you could even just label as double/single membrane vesicle directly on the image?
10. A recent paper by S.M Dolliver et al 2022 PLoS Pathogens (PMID 36534661) published some interesting work on SG formation in SARS-2 infection, with some affirming findings (absence of SG formation) and some conflicting findings (sharp reduction in G3BP in infected cells, and antiviral functions of G3BP1) to your results herein. It would be important for the authors update the text to include a discussion of this. For example, why have others observed the reduction in G3BP1 in SARS-2 infected cells but not in your study? Line 287 should also be updated to include this work and reference.

Overall, very nice work!

Kind regards,

Dr. Christina Guzzo
University of Toronto Scarborough

Reviewer #3 (Remarks to the Author):

In this manuscript, Murigneux and colleagues analyze host factors associated with SARS-CoV-2 virions using two different purification methods and liquid chromatograph-coupled mass spectrometry. G3BP1 and G3BP2, two major proteins involved in stress granule formation, were

identified as the most enriched host factors in virions. Evaluation of their role during the life cycle of SARS-CoV-2 allowed the authors to propose a model in which G3BP1/2 contribute to the formation of cytoplasmic membrane structures that might facilitate virion assembly. The authors present a well-structured and controlled study in which they reveal the previously unidentified host factor content of SARS-CoV-2 particles. While this is certainly a resource for the field, it appears that the inclusion of host factors in particles is somewhat arbitrary, as other candidates besides G3BP1/2 have no or opposite impact on viral replication. Furthermore, the key question of the importance of G3BP1/2 in particles is not addressed. The manuscript focuses on the cellular role of G3BP1/2 during viral infection, confirming their previously described interaction with N, and proposing a new proviral function. The model for the role of G3BP1/2 in particle assembly is very interesting but the conclusions remain preliminary at this stage and require additional ultrastructural evidence to be supported (see my specific comment below).

- A549-ACE2 cells are a valuable tool but show very slow particle release kinetics compared to the Calu-3 human airway epithelial cells, which in addition also express TMPRSS2 that is important for S maturation. It is therefore important to demonstrate that the incorporation of G3BP1 and G3BP2 into particles and their effect on particle assembly is not cell type dependent.
- Is the silencing of G3BP1/2 also proviral in other relevant cell lines, e.g. Calu-3 cells?
- The presence of G3BP1/2 into purified virus particles should be validated by immunogold labeling along with S or N.
- The bioinformatics analysis comparing the interactions of the virion-associated host candidates identified in this study with other host proteins identified from cells or in different viruses (Fig. 2b and Supp. Fig. 1c) is interesting but does not add much. It could be shortened.
- Additional staining for ORF7a and for nsp3, as a viral protein that is not incorporated into the particles, would strengthen the results regarding the quality of the purified fraction.
- In general, input/cell lysates and purified fractions should be loaded onto the same SDS-PAGE gel (Fig 3e, 4c, Supp Fig 2a, Supp Fig 3c). On the same line, the panels for G3BP1, G3BP2, and GAPDH staining are absent.
- The authors mention the absence of exosome contamination. Validation of representative marker molecules in virion preparations by Western blot would support this point.
- Virus preparations shown in Fig.1b and in Fig.3b, e and f differ regarding to the levels of S2. Does ACE2 affinity capture enrich particles carrying uncleaved S? What is the infectivity level (TCID50) of these particles compared to those passed through the sucrose cushion?
- The effect observed upon depletion of G3BP1/2 is moderate, on average twofold and is not reported in TCID50/ml units. Viability assays in the presence and absence of viral infection should be provided for the siRNA and Cas9 transient experiments to exclude an artifactual effect of the method.
- The method used to deplete G3BP1 and G3BP2 appears to impact intracellular gRNA levels. In transient Cas9 assays (Fig. 3b), levels are reduced by 25% while unaffected by silencing. This undermines the main argument that G3BP1 does not impact replication but assembly.
- A549 stable G3BP1/2 double knockout cells seem to be viable (<https://www.biorxiv.org/content/10.1101/2021.04.26.441141v2>). Such cells would help clarifying the effects.
- The fact that binding between N and G3BP1/2 depends on an interaction with RNA in cell extracts is interesting. How were the virion samples treated with RNase? Were virions ruptured first? This information is missing from the method section.
- Fig.6d, representative Western blot image: the level of S in the total lysate is visibly reduced. This is misleading. In addition, total lysates and purified fractions should be analyzed on the same gel.
- Supp Fig 4 shows the degree of colocalization between G3BP1 and N in infected cells. This type of analysis is not provided for the expansion microscopy. What is the degree of colocalization between N and S and N and M in ctrl vs. G3BP1/2 kd cells? This information would provide evidence to support the model. Can G3BP1/2 interact directly with S?
- Expansion microscopy is an impressive method for improving resolution. However, when it comes to visualizing differences affecting viral assembly sites, analysis at ultrastructural level is more relevant. Electron microscopy combined with immunogold labeling of S and N would support the observations and the proposed model more convincingly. This would additionally exclude the possibility that the substructures observed in expansion microscopy are in fact intracellular accumulation of viral particles.

We would like to thank the reviewers for the time and effort in reviewing our manuscript, as well as for the constructive feedback. We have carefully addressed all the comments and suggestions, and we believe that the revised manuscript now substantially improves the quality and clarity of our research.

Attached, please find the revised version of our manuscript: one Word document corresponding to the final version of the manuscript (Murigneux et al, final version) and another one with the comparison to the previous version (Comparisons Manuscripts Murigneux et al) along with a detailed response to each of the reviewers' comments.

We hope that the revisions we have made have adequately addressed the reviewers' concerns.

Thank you once again for your consideration. We look forward to hearing from you and hope that our revised manuscript will be favorably reviewed.

REVIEWER COMMENTS

Reviewer #1 (Remarks to the Author):

1. The limitations of virion proteomic analysis (see below) are not described and contextualized and I think it is important for the reader to understand the limitations of the datasets that they are looking at and help in the process of making educated guesses in the selection of candidates. I understand that the data looks good for the standards of viral particle analysis, but still, many of the discovered proteins will likely be bystanders.

- To make clearer the limitations of our study we have added the following sentence in the result section (p8, line 192-193): “Virion assembly inherently incorporates a portion of the cytoplasm into the viral particles, potentially resulting in the passive inclusion of abundant cellular proteins.”
- we have also added the following paragraph to the Discussion (p17, line 402-412):

“Distinguishing between proteins passively incorporated during virion assembly and actively recruited with functional relevance can be challenging. As described by Garcia-Morena et al., we noticed a correlation between the LC-MS signal intensity of factors associated with virions and their protein abundances in A549 cells ($R=0.42$; Supplementary Fig. 7a). However, when considering the degree of enrichment of factors in virion-associated fraction as compared to control fraction from mock-infected cells, we observed a significant reduction in the correlation with protein abundance ($R=0.13$), in particular for SARS-CoV-2 RBP ($R=-0.0093$) (Supplementary Fig. 7a), suggesting that the degree of enrichment is a better indication of an active association with virions⁷¹. **Isolating virions from other extracellular vesicles is also challenging.** Consequently, virions isolated on sucrose cushion were further purified based on their affinity for ACE2 receptor”.

2. Analysis of the proteome of the viral particles. I think overall, the authors have done a good job when using the sucrose gradient approach, but I don't understand some of the decisions taken. For example, authors applied a sound approach using well-established label

free quantification and estimation of the false discovery rates (FDRs). However, they take the arbitrary decision of following a semi-quantitative approach in which only proteins identified in the viral particles are considered. I think this makes little sense from the proteomic point of view. When considering a given protein in two samples, identification and quantification is a matter of signal and noise as well as peptide crowding at a given point of the chromatogram. Black and white situations are not necessarily the best, because they can represent cases in which one protein is detected at trace level in condition A and is absent in condition B. This would be considered as an optimal candidate although it is actually not a good one. Hence, the peptide intensity should be factored in some way. The quantitative approach does that (particularly if applying noise imputation and FDR correction) by generating ratios that are dependent on intensity levels in both samples. I understand authors are reluctant because of the intrinsic properties and limitations of the analysis of viral particles, however, these limitations can be counteracted in other ways as described below. However, I don't think that the strategy of discarding the quantitative approach and replacing it by a semi-quantitative approach improves the data in any way.

- In our initial analysis, we adopted a stringent approach by considering proteins present exclusively in the supernatant of infected cells. This was done to minimize the inclusion of potential contaminants associated with extracellular vesicles rather than virions. However, we recognize the validity of the reviewer's comment that this approach omitted the quantitative aspect of the analysis.
- In response to this suggestion, we re-evaluated our data using noise imputation and FDR correction. This revised approach led to the identification of 284 proteins (with adjusted p-value ≤ 0.05 and FC > 1.5) associated with virions produced from A549 cells, including the 137 proteins that were initially identified in our previous analysis. 21 proteins were excluded from our prior analysis, while an additional 147 new virion-associated factors were revealed (Fig. 1f and Fig 2). Notably, these newly identified factors belong to the same Gene Ontology (GO) categories as those highlighted in our initial analysis.

We applied similar quantitative analyses to both virion-associated factors enriched from Calu-3 cells and those enriched in the viral fraction isolated via ACE2-affinity capture. In both cases we selected statistically enriched factors based on a p-value of $p \leq 0.05$ (and FC > 1.5). The change from using the more stringent adjusted p-value is due to the challenge of generating sufficient virions for MS analysis either from Calu-3 cells that grow very slowly combined with a high background noise in controlled conditions or from A549-ACE2 cells after a two steps of virion purification (sucrose cushion and ACE2-affinity capture).

3. The problem of viral particle proteomics is that virions assemble inside the cell and in the process will inevitably take a portion of the cytosol (or compartment where this happens) that can perfectly contain protein with no role in infection. Indeed, comparison of the intensity of proteins in virions purified in previous studies correlate very well with that of the cell proteome.

Therefore, there is different ways to benchmark the results that are not considered here, including whole proteome, generation of particles in other cell lines, etc.

- We agree that one limitation of virion proteomic studies is that factors associated with virions can be incorporated non-specifically, especially if they are abundant in

infected cells. However, we anticipated that functionally relevant factors are more likely actively recruited to virion assembly sites through interactions with virion components (structural protein or gRNA). Subsequent bioinformatic analyses were thus performed to intersect our data with other relevant studies and investigate their interactions with virion components (Fig-2). In line with this, a very recent study by Garcia-Moreno et al. (Biorxiv 2023) reported that the intensities of factors associated with HIV RNA within HIV-1 virions correlate with their cellular abundance. Yet, the enrichment of HIV RNA binding factors identified in virions, when compared to the whole-cell proteome, exhibited an inverse correlation with protein abundance. To explore this aspect further, we plotted the signal intensities (LFQ) of factors associated with virions against their abundance in infected A549-ACE2 cells. We confirmed a correlation between the signal intensity (LFQ) of factors associated with virions and their abundance (see Supplementary Fig. 7). However, when considering the enrichment (FC) of factors associated with virions compared to preparations from mock-infected cells, there was a significant reduction in the correlation with protein abundances. This suggests specificity in the association of these factors with virions (see Supplementary Fig. 7). Additionally, proteins known to interact with SARS-CoV-2 RNA exhibited an inverse correlation with cellular protein abundance, indicating a specific recruitment pattern for these proteins (Supplementary Fig. 7). This point is now added in the discussion (Page 17, Line 402-419)

- In addition, as suggested by the reviewer 1, we benchmarked our data by analyzing the proteome content of SARS-CoV-2 virions produced from Calu-3 cells, another pulmonary cell model naturally expressing ACE2 and TMPRSS2. 120 factors were statistically enriched in virions produced from Calu-3 cells ($p \leq 0.05$ and $FC \geq 1.5$) and 48 factors were identified in common between the two conditions (including G3BP1 and G3BP2). These data are now included in Figs 1, and 2, Supplementary Figs 1, 2 and 3a). Fig 2 shows that factors specifically enriched in virions produced from Calu-3 cells (triangles and polygons) essentially belong to the same biological pathways and molecular function (ribosomes and translation, ubiquitin and proteasome pathway, RNA metabolism) as those that interacted closely with virion-associated factors from A549 cells (diamonds and polygons).

Authors try to overcome this by applying a second way of isolation of virions, using biotinylated ACE2. Unfortunately, this experiment seems tremendously inefficient and is not well controlled (for example capture using the biotinylated ACE with the preparations from uninfected cells). Therefore, the results aren't convincing and can account only for a minority of the proteins identified in the virions. The experiment is not particularly useful and raises questions about isolation efficiency and missing controls to account for biases and I would personally not trust the data provided.

- We respectfully disagree with the reviewer's comment on the inefficiency of the experiment. Although less material was recovered there was a gain in specificity and information, as it emphasizes factors associated with intact SARS-CoV-2 virions decorated with spike proteins at the virion surface. Limiting exosome contamination is a major challenge in virion isolation strategies, and including this additional ACE2 purification step led to a gain in specificity and validation of promising candidates.
- Regarding an apparent lack of appropriate controls, as we previously stated, ACE2 affinity capture distinguishes between factors associated with viral particles displaying spike proteins on their surface from factors associated with contaminating

extracellular vesicles. Importantly, our analysis cross-referenced the ACE2 affinity capture factors to those factors enriched in sucrose cushion preparations from infected cells **compared to non-infected cells** (Fig 1 and 2). Factors present in the preparations of non-infected cells, but not enriched in virions subsequently used for ACE2 affinity capture were thus automatically excluded, whether they were captured with ACE2 or not.

- Furthermore, we controlled the specificity of ACE2-biotin capture by using biotin to eliminate non-specific binding of factors that were enriched in viral preparations, but not associated with virions bearing spike protein.
- Also, we performed an additional ACE-2 affinity capture step using ultracentrifuged supernatants on sucrose cushions from both **uninfected and infected cells**. We found 131 host factors enriched in virions isolated by ACE2 capture compared to the biotin control, including 91 in common with the 284 virion-associated factors purified via sucrose cushion ($q < 0.05$ and $FC > 1.5$; $n=4$) and 50 in common with the 71 factors identified in other ACE2 affinity capture ($p < 0.05$ and $FC > 1.5$; $n=3$). 14 factors were detected in ACE2 capture from mock-infected preparations, 6 of them were enriched compared to biotin control ($FC > 1.5$). Two of these enriched factors were in common with the 384 virion-associated factors defined via sucrose cushion, and none of them overlapped with the 71 factors identified following ACE2 affinity capture, or in any other ACE2-affinity capture experiments ($p < 0.05$ and $FC > 1.5$; $n=3$) (Supplementary Dataset 3).
- Regarding this additional experiment, the data of the ACE2 affinity capture on the ultracentrifuged supernatant of infected cells was added to Fig 3 and Supplementary Fig. 3a, and regarding the control using supernatant of non-infected cells, the following sentence was added (p10, line 227-230): “None of these proteins were detected in ACE2 capture from non-infected supernatant in a control experiment (Supplementary Fig. 3b and Supplementary Dataset 1).”

Another limitation is that proteins can be soluble or trapped at the surface of the virion. Authors tried to minimize this by using an amicon purification step, that I am sure helped but will not eliminate proteins forming large (>100KDa) complexes or attached to the surface of the virion. Trypsin digestion of the sample before the gradient separation would help as the envelope is supposed to protect intravirion proteins. This latter point might be difficult to counteract at this point, but should be at least used to determine if the proteins characterized in Figure 4a are inside the viral particle.

- Amicon purification helped to eliminate a lot of background, but we cannot exclude that proteins associated in large complexes might not be eliminated by amicon purification. Of note, 67 of the 356 proteins (18,8%) identified in association with SARS-CoV-2 virions from A549 and Calu-3 cells have a molecular weight above 100 KDa.
- We thank the reviewer for suggesting tryptic digestion. To confirm the presence of proteins of interest within viral particles, we treated isolated virions for 15 min with subtilisin, stopped digestion by inactivating the trypsin-like protease and isolated virions by ultracentrifugation, as described in (Perez-Caballero, Cell 2009). As expected, this resulted in the digestion of the extracellular domain of Spike protein,

but had no effect on N protein. Furthermore, subtilisin treatment of virions produced by A549 and Calu-3 cells did not impact the association of G3BP1, G3BP2, YBX-1, IGF2BP1, IGF2BP3 and PABPC1 with SARS-CoV-2 virions.

4. Authors detect initiation factors and ribosomes inside the viral particles. This can be a consequence of defective exit of translation before packaging or can be a mechanism to 'label' the viral RNA for translation in the newly infected cell. Analysis of the coverage (which ribosomal and eIF proteins are detected) and intensities of these proteins would shed light on this idea. Are they bystanders or functionally necessary (I understand it is not the focus of the paper, so no experimental validation is required).

- The table below includes the ribosomal and eIF proteins detected in virions produced from both A549 and Calu-3 cells isolated via ultracentrifugation on sucrose cushion and via ACE2 affinity capture, along with the coverage, intensities and fold changes as compared to control, as requested. These data are also available in Supplementary Dataset 1.

A549-ACE2		LFQ with imputation									
	mean Crispr Score	% coverage	SARS_1_imputed	SARS_2_imputed	SARS_3_imputed	SARS_4_imputed	CTL_1_imputed	CTL_2_imputed	CTL_3_imputed	CTL_4_imputed	
RPL5	-3,1951125	41,4	77144,0	262560,0	201530,0	166310,0	953,6	10284,8	5259,3	2803,6	
RPS2		22,2	78303,0	250470,0	149720,0	98150,0	1706,5	4543,5	5749,2	1491,5	
RPLP0	-2,6073	36,6	149130,0	383060,0	280180,0	155060,0	2025,2	26707,0	6748,3	1700,1	
RPS11	-2,931925	43	77148,0	227830,0	152070,0	99411,0	3128,2	5930,3	3980,9	2283,3	
RPS4X	-3,043319444	48,3	118770,0	315620,0	180560,0	93570,0	1206,4	70552,0	6960,7	1603,7	
RPS3A	-3,715444444	45,1	46729,0	189750,0	123150,0	70905,0	1731,2	9089,6	5864,9	1824,4	
RPL7		30,2	85704,0	70487,0	79248,0	45468,0	2128,0	2928,2	6403,3	1381,4	
RPL7A	-3,448611111	35	85891,0	158310,0	119620,0	78971,0	1981,6	11288,0	12406,0	1430,9	
RPL3	-3,2748625	36,2	35251,0	99658,0	109620,0	87816,0	1911,6	11414,0	8310,2	732,4	
RPS16	-2,348222222	41,8	118350,0	105940,0	111650,0	59013,0	2188,4	9377,7	6727,5	2882,8	
RPS18	-2,3503125	42,8	143780,0	123890,0	113810,0	73761,0	4377,6	10634,0	7601,2	2563,1	
RPS24	-0,4209875	30	64314,0	135500,0	93309,0	53836,0	955,1	9729,6	6890,3	4286,1	
RPL18	-3,718475	45,9	69701,0	129110,0	108860,0	69910,0	1477,0	36378,0	5218,3	1583,3	
RPL27	-2,247055556	39,7	40429,0	125540,0	72445,0	59228,0	1684,9	8674,9	5711,4	2004,5	
RPL23	-3,024875	25	5997,3	196950,0	140280,0	243450,0	1186,2	11859,4	7816,2	3111,7	
RPS9	-3,16215625	40,2	54004,0	140880,0	73147,0	52475,0	2489,8	10215,0	6886,6	1589,2	
RPS14	-3,1842375	36,4	104030,0	94257,1	47317,0	32832,0	1776,5	8819,4	7720,0	1405,7	
RPS26		42,6	84247,0	164550,0	163680,0	64154,0	4028,0	12916,7	10773,0	3218,4	
A549-ACE2											
RPL4	-2,6734375	28,3	37294,0	155340,0	58513,0	44245,0	1591,0	7490,2	7369,1	2268,6	
RPL17	-2,559930556	34,2	39479,0	96630,1	54694,0	39648,0	2426,4	7657,2	8742,7	999,6	
RPL14	-2,9189625	22,3	66949,0	70914,0	15235,6	65716,0	1311,6	6168,0	9133,3	1912,7	
RPS13	-3,1277875	32,5	51315,0	93602,0	51960,0	39529,0	5466,2	8452,4	8082,9	812,7	
RPL30	-2,5279375	67	20786,0	147990,0	60807,0	42132,0	1868,8	9749,2	8310,6	2460,4	
RPS8	-3,44021875	43,8	107910,0	86914,0	102020,0	72541,0	4048,3	33997,0	8332,9	3126,6	
RPL13A	-3,2554375	15,3	60486,0	16044,9	68280,0	49213,0	3067,4	4336,8	13380,0	1073,8	
RPS17		62,2	24879,0	70173,0	54801,0	19883,0	3966,4	6315,9	5794,6	1152,6	
RPL21		9,4	25800,0	68763,0	54275,0	32128,0	2137,2	11040,5	6724,5	1756,1	
RPS5	-2,6136	26,5	53826,0	92245,0	10556,3	36372,0	2327,2	6256,3	7789,8	1631,4	
RPL10	-3,7581375	42,1	9329,4	60979,0	63811,0	27578,0	1929,6	3028,0	10154,3	1747,0	
RPL10A	-2,1452125	36,9	34733,0	107450,0	39518,0	22886,0	2707,0	11238,0	7811,1	1513,0	
RPL35A	-3,204025	28,2	11061,5	64430,0	24245,0	40126,0	1601,9	7502,4	5534,6	1257,3	
RPL8	-3,998525	9,7	42276,0	63297,0	51353,0	57954,0	3101,3	6710,1	15325,6	3597,1	
RPS20	-2,16655	22,7	57400,0	59860,0	33864,0	20878,0	3764,4	6230,5	9786,8	1527,7	
RPS6	-3,310575	27,7	51994,0	35132,0	64254,0	7943,7	1241,6	6890,5	14736,3	1873,7	
RPL18A		36,4	5544,6	54800,0	77980,0	80012,0	2458,1	22530,0	4656,8	1965,2	
RPL6	-3,065763889	22,9	5259,9	78959,0	30232,0	21850,0	2827,7	11386,3	9258,0	1183,5	
RPL15	-3,503125	32,8	31876,0	81264,0	39787,0	5766,0	4023,3	7702,1	5956,0	4309,0	
EIF4A1		39,7	80783,0	184940,0	144720,0	69749,0	3175,5	15777,0	7862,4	1536,5	
EIF5A	-2,0232	46,1	63482,0	127460,0	88564,0	22061,0	1679,5	3705,7	5200,0	2243,8	
EIF3C		10	38369,0	84530,0	47541,0	35280,0	2109,0	6708,4	9679,9	3395,2	
EIF3L	-1,2849625	9,7	55191,0	37613,0	44603,0	30268,0	1239,0	11532,5	10942,1	2227,1	
EIF3A	-2,48755	9,7	11776,0	30838,0	31825,0	14471,0	2130,4	3139,3	6647,6	2686,5	
EIF2S3	-2,664236111	16,9	16807,0	24556,0	15018,0	7072,6	4254,2	4270,4	3303,6	1613,6	

		LFQ with imputation								
		mean	% coverage	SARS_3_imp	SARS_4_imp	SARS_5_imp	CTL_3_imp	CTL_4_imp	CTL_5_imp	
Calu-3	RPS13	-3,1277875	43,7	202940,0	123650,0	113810,0	84557,0	7618,3	7912,7	
	RPS5	-2,6136	39,2	46864,0	84074,9	61854,0	6931,1	19804,0	11334,5	
	RPS10	-2,883775	35,2	21106,9	48181,0	34098,0	8912,4	9663,4	17370,5	
	RPL22	-1,6466625	55,5	14944,0	100030,0	57810,0	9925,2	45051,0	26350,2	
	RPS8	-3,44021875	60,6	488430,0	787720,0	334610,0	232990,0	543700,0	249390,0	
	RPL7A	-3,448611111	47	232330,0	262690,0	190530,0	153390,0	134640,0	153640,0	
	EIF3L	-1,2849625	11,6	24912,0	164040,0	147510,0	8869,0	16422,0	22628,2	
	EIF3I	-2,244875	23,4	21896,8	118850,0	127660,0	9901,6	13559,8	19811,2	
	EIF3M	-1,1337	15,5	45140,0	60662,0	60750,0	9391,1	29555,0	10213,3	
	EIF4A3	-4,3186625	24,6	22063,7	111110,0	99788,0	13669,7	21326,6	21731,3	
	EIF2S1	-3,9290125	27,3	23850,2	60720,0	59451,0	12110,1	13540,9	16421,4	
	EIF4G1	-1,793125	13	59047,0	59472,0	53047,0	20764,4	16335,6	31398,0	
	EIF3D	-3,232361111	16,2	22573,6	44831,0	40089,0	13272,4	12332,6	18750,1	
	EIF2S2	-2,3346375	12,6	17398,1	22791,0	25348,0	13010,7	9795,9	16848,4	
	EIF4A1		44,6	301890,0	555570,0	532270,0	192550,0	313230,0	320660,0	

		LFQ with imputation									
		mean	% coverage	SARS_1_imputed	SARS_2_imputed	SARS_3_imputed	SARS_4_imputed	CTL_1_imputed	CTL_2_imputed	CTL_3_imputed	CTL_4_imputed
Capture	RPS11	-2,931925	43	148335,0	157141,0	310127,1	111772,1	13783,9	1836,7	1650,9	1041,7
	RPS3A	-3,715444444	45,1	62505,7	105107,1	233117,9	51833,6	31123,1	1706,8	1677,2	2117,7
	RPS2		22,2	67495,3	101688,7	199219,6	37037,5	73588,9	3126,2	820,1	2148,3
	RPS26		42,6	68078,0	114620,1	126607,2	4006,4	14666,1	2640,7	1241,9	2391,0
	RPL7		30,2	144398,9	86271,7	127443,7	4369,0	66584,6	1829,1	1250,6	1359,0
	RPL8	-3,998525	9,7	81685,9	146140,9	267967,8	5017,7	55677,2	2752,1	6179,2	551,3
	RPS13	-3,1277875	32,5	73349,5	142786,5	220145,8	3184,6	36700,2	1964,0	4846,4	1068,0
	RPL7A	-3,448611111	35	258337,9	76926,9	227073,5	69340,0	62848,9	25253,6	6519,3	1818,8
	RPL14	-2,9189625	22,3	39087,0	95512,4	85830,3	4632,6	15335,6	4187,0	1015,7	2617,9
	RPL3	-3,2748625	36,2	146506,0	60280,3	137836,6	2774,9	51966,7	5376,3	5633,4	452,4
	RPL15	-3,503125	32,8	22619,8	117451,0	204097,6	2045,9	18871,2	8859,4	3737,9	568,9
	RPS4X	-3,043319444	48,3	120552,3	148499,6	306409,2	4451,0	75281,1	7818,1	40744,0	1202,6

- Enriched factors included translation initiation factor (eIF2S3), Cap-dependent translation initiation factors (eIF4A1, eIF5A), as well as elongation (eIF3A, eIF3C, eIF3L, EEF1G, EEF2) factors. Ribosomal proteins belonged to the small (RPS10, RPS11, RPS13, RPS14, RPS16, RPS17, RPS18, RPS2, RPS20, RPS24, RPS26, RPS3A, RPS4X, RPS5, RPS6, RPS8, RPS9), as well to the large (RPL10, RPL10A, RPL13A, RPL14, RPL15, RPL17, RPL18, RPL18A, RPL21, RPL22, RPL23, RPL27, RPL3, RPL30, RPL35A, RPL4, RPL5, RPL6, RPL7, RPL7A, RPL8, RPLP0) ribosomal subunits. These ribosomal proteins are involved in various stages of translation, from initiation to elongation and termination, and some of them are necessary to maintain the integrity and activity of the ribosome.
- We cross-referenced this list of ribosomal and EIF proteins with a large-scale CRISPR-CAS9 screen (Wang et al., Science 2015, PMID: 26472758), confirming that all the detected EIF and ribosomal proteins are essential genes for human cells. However, it is currently challenging to determine whether the presence of these proteins in virions results from a defective exit of translation of packaged viral RNA or serves as a mechanism to label viral RNA for translation upon infection and we prefer not to elaborate on what role they might play.

5. There is a discussion about whether G3BP1/2 would play its role in infection by producing phase transition or not. This however isn't clear as G3BP1/2 are active RBPs with role in infection in their 'soluble' form.

The fact that other SG proteins are not in the virions suggest that the role of G3BP1/2 are likely to be in their soluble form. Analysis of TIA1, TIAR, etc in the virions and IPs (Figure 6 – intracellular vs virions) would help to determine whether is a lack of incorporation or just that the sensitivity is not enough to detect them. Also the fold change of these proteins in the proteomic analysis should be provided as they might be enriched in a not significant manner or disqualify for the on/off criteria that the authors applied.

- We acknowledge the uncertainty regarding the role of phase separation in the function of G3BP1/2 in virion production, as mentioned in our Discussion (p20, line 474-475): “Whether G3BP1/2 carry out this function by mediating phase separation condensates remains to be determined.” However, we believe that this hypothesis cannot be ruled out due to the absence of other stress granule proteins in virions. Notably, TIA-1 and other SG are not essential for stress granule formation (Protter and Parker, Trends Cell Biol 2016). Recent studies have shown that purified G3BP proteins and RNA can spontaneously reorganize into condensates, acting as a scaffold for recruiting other RNA-binding proteins (RBPs). While other SG proteins participate by weakening or strengthening G3BP/RNA condensates, the interactions among G3BP proteins and RNA molecules are sufficient to drive phase separation (Guillen-Boixet et al., Cell 2020; Yang et al., Cell 2020).
- Moreover, several studies reported that N and RNA, as well as N and G3BP1 can form condensates *in vitro* and in cells (Cascarina et al, 2020; Caubuk et al, 2021; Inerman et al, 2020; Lu et al, 2020; Seim et al, 2022). A recent study from Yang et al further showed that addition of purified G3BP1 favored N-RNA condensate formation *in vitro* (Yang et al, Biorxiv)
- As requested, the presence of TIA1, TIAR and CAPRIN was analyzed in the viral particles (Figs. 3e and 3f, Supplementary Fig. 3b, Supplementary dataset 1) and in the IPs by western-blotting but were not detected (Figure 6e).
- The fold change in the proteomic analysis of TIA1, TIAR, CAPRIN or UBAP2L cannot be provided as they were not detected in any ultracentrifuge preparations.

6. Phenotypes of G3BP1/2, IGF2BP1, PABPC1. These are all very central to RNA metabolism. Therefore, it would not be surprising that their effects in infection are actually indirect (as each regulates hundreds/thousands of cellular transcripts). I think these experiments should be better characterized. For example, do the overexpression of these proteins affect infection? Can the phenotype be rescued? What is the effect of the KD in the cell gene expression (any available RNAseq in KD cells? eCLIP?)?

- We agree with the reviewer that some of these RNA-binding proteins (RBPs) could participate in SARS-CoV-2 replication by regulating host genes.
- Through iCLIP experiments, Nabeel-Shah et al. reported that interactions of N with G3BP1 rewired the G3BP1 mRNA binding profile and protein interactome is also affected (Kruze et al 2021, Yang et al Biorxiv 2023), as mentioned in our manuscript (page 17, line 421-423): “The interaction of SARS-CoV-2 N and nsp1 with G3BP1 was reported to reduce SGs assembly, alter G3BP1 mRNA-binding profile, modify its interactome and dampen host immune responses^{12,61–63,65,67,70,73,74}.”
- G3BP1/2 associate with N and viral RNA in the cells and with N within virions. These proteins also favor the association of N with S in ERGIC derived granules that are likely sites of virion assembly, and favor the production and release of virions at a late stage of replication (post-translational). We cannot exclude the possibility that G3BP1/2 phenotype might be indirect. However, the direct or indirect functions of multifunctional proteins in cellular mRNA metabolism does not exclude another potential role in late stages of virion production, such as stabilizing viral RNA, preventing its translation, or favoring its packaging. Moreover, it's essential to note

that the majority of RNA-binding proteins interacting with SARS-CoV-2 RNA (503 factors) were not identified as SARS-CoV-2 virion-associated factors, suggesting some specificity in the incorporation of these particular factors.

- As IGF2BP1 and PABPC1 were not the primary focus of this study, we did not conduct additional experiments related to these proteins. However, we have performed G3BP2-HA overexpression experiments as per the reviewer's request. Fig. 4d indicates that while G3BP2-HA overexpression modestly increases (1.2 fold) the index release of viral RNA in CTL cells, it fully restored the decrease in release index induced by G3BP1/2 double KO (Fig. 4f).

7. G3BPs are antiviral in several other studies. What are the experimental differences with the present study? How these results can be conciliated?

- Contradictory effects have been observed regarding the role of G3BP proteins in SARS-CoV-2 replication: Zheng et al and Liu et al suggested an antiviral function of SARS-CoV-2, whereas Ciccocanti et al and Krüze et al suggested a proviral function.
- As discussed in the manuscript (p 18, line 437-439), this apparent contradiction may be explained by differences in experimental conditions. Several cell models were used, (HeLa-ACE2, A549 transfected with ACE2 expressing plasmid before infection, Vero cells). The conditions of infection were not the same: different MOI (MOI=0.1 or 0.001 or not specified), different time points after infection: (24 hpi, 36 hpi, 48 hpi) and SARS-CoV-2 replication was measured with different assays (intracellular dsRNA, intracellular viral RNA level, extracellular viral RNA level, intracellular N level).
- Our data show that down regulation (KD and KO) of both proteins in two relevant cellular models (A549-ACE2 and Calu-3 cells) decreased the release of virions, assessed at the RNA and protein (N and S) levels at 10 hpi, This effect likely impacts on the overall replication of SARS-CoV-2, as observed at 48 hpi.
- Assessing the impact of G3BP proteins during a single round of infection (10 hpi, high MOI), rather than after multiple cycles of replication (48 hpi, low MOI), enabled a clearer understanding of the role of these proteins at individual stages of the SARS-CoV-2 replication cycle.
- Moreover, G3BP proteins are multifunctional proteins involved in SG assembly, but also RNA metabolism and immune response. Assessing the impact of G3BP1/2 KD and KO at 10 hpi enabled monitoring the effect of G3BP1/2 before SG formation and IFN response (p 18, line 431-442).
- Importantly, we and others have shown that G3BP1 and G3BP2 are two proteins with redundant functions, and that the KD of one increases the level of the other (Supplementary Fig 3a and Cristea et al., 2010; Kedersha et al., 2016; Matsuki et al., 2013). Therefore, a main difference with other studies is that we knocked-down expression of both G3BP1 and G3BP2 to prevent compensatory effect of one protein due to loss of the other.

8. Figure 4. What about viral titre, plaque morphology, etc ? G3BPs should then reduce the

viral titre, and may have additional effects in a second round of infection (reflected in plaque number and size).

- Virus titers were quantified by challenging A549-ACE2 cells with serial dilutions of virion-associated fraction in 96 well plates. The 50% Tissue Culture Infectious Dose (TCID₅₀/mL) was calculated by examining cytopathic effects on the infected cells (Figs 4d and Supplementary 4h).
- Fig 4d indicates that the viral titer (TCID₅₀/mL) of virion-associated fraction from G3BP1/2 KD A549-ACE2 and Calu-3 cells is decreased by approximately 70% as compared to their respective controls.
- Fig 4g further indicates that the infectivity (viral titer normalized to the quantity of virion assessed by Elisa quantification of N) is decreased by approximately 50% in G3BP1/2 KD cells as compared to control cells.

9. Figure 5. What is the localisation of N and other viral proteins in G3BPs KD? Controls of the antibodies in the mock cell and KD cells should be included. The analysis of the microscopy data should be more systematic, providing Pearson correlation of the different channels and quantification of N in WT and KD.

- We have included the control conditions (mock-infected) for N and S antibodies in Fig. 6a. New data were acquired so that identical settings were used between mock- and SARS-CoV-2-infected CTL and KD cells in Fig. 6a.
- We did not observe a change in the overall localization of N and S due to G3BP1/2 KD (Fig. 6a). Pearson correlations were included in Supplementary Fig 6.
- Fig 6c shows that quantification of cells harboring N foci are identical in siCTL and siG3BP1/2 conditions. In line with this, we quantified the number of N foci/cells (90 cells in 3 independent experiments). The figure below indicates that the number of N foci/cells are similar in G3BP1/2 KD cells and in control cells. Importantly, quantification of the number of cells harboring N foci with S colocalization is decreased by two-fold in G3BP1/2 KD cells compared to control (CTL) cells (Fig 6c).

10. Figure 6d shows that KD of G3BPs decreases N levels in both whole cell proteome, cytosol (particularly) and membranes. WB does not reflect the quantification showed in the right panel, in which cytosol quantification is not included. The results showed disagree with

the ELISA results presented previously and suggests that KD affect N to some extent. Indeed, a slight decrease of N is also observed in inputs of panel e.

- We apologize for the confusion, but for technical reasons the cell fractionation experiments were performed at 24hpi, after multiple rounds of SARS-CoV-2 replication. The decrease of N and S in the total cell lysate and in the cytoplasmic fraction in Fig. 6d is a consequence of the decrease of SARS-CoV-2 replication in G3BP KD cells, as compared to control and was expected as mentioned p15, line 369-371 of the manuscript. Of note, the decrease of N appeared more pronounced in the cytosolic fraction but the loading of G3BP1/2 KD sample was a bit reduced as compared to control (see GAPDH).
- Therefore, this observation is consistent with a proviral function for G3BP1/2 in supporting SARS-CoV-2 replication. Please note that this experiment should not be compared to N Elisa quantification (that has been moved to Supplementary Fig 4e due to space constraints), as the N Elisa samples were analyzed at 10hpi (after a single round of replication), a time point at which we did not observe any effect on cell-associated viral protein levels, consistent with western-blot quantifications.
- To compare the recruitment of N to S-containing membranes in G3BP1/2 KD cells and CTL cells, we normalized the level of N to the level of S in both the total cell lysate and the membrane fraction of Fig 6d. The quantification on the right of the western-blot reflects the ratio of N to S levels in these fractions. We acknowledge that the previous labeling of the quantification may have been confusing and have since been updated. This ratio N/S could not be calculated for the cytosolic fraction as S is not present in the cytosolic fraction.
- We are uncertain about the reviewer's comment regarding panel e. In panel e, we did not perform any Knock-down. The inputs in panel 6e represented the same viral production with and without RNase treatment. The possible slight difference observed was due to variations in loading and was not related to the effect of G3BP KD (see GAPDH band intensities). We have replaced the gel in the previous version of the manuscript to include CAPRIN, TIA-1, and TIAR controls, as requested.

11. Figure 6e. The observed differences may well have a different explanation, which is compaction. Viral RNA might be more accessible to RNases when is not or partially associated to N than when it is fully compacted and recovered into viral particles. Analysis of the RNA after RNase in both conditions should be provided.

- RNase treatment was performed after virion lysis (in 50 mM Tris pH 8, 150 mM NaCl, 0.25% Triton X-100) and before N Immunoprecipitation from both cell- and virion-associated fractions.
- Nevertheless, to confirm the efficacy of the RNase treatment, quantifications of viral gRNA levels from cell and virion lysates with and without RNase treatment were performed in parallel to the IP experiments. In untreated samples, gRNA levels were detectable (Ct between 10.23 and 13.03 in cell lysates and between 13.81 and 16.71 Ct in virion lysates). However, in RNase-treated samples, we could not detect any trace of viral RNA (Ct > 35) in either cell or virion lysates. Data could not be normalized to any control in RNase-treated samples. A graphical representation of $2^{-\Delta\Delta Ct}$ values is provided for the reviewer. However, we simply mentioned (p 44, line 1085-1086)

that “The efficiency of RNase treatment was verified in parallel on gRNA level by RT-qPCR” in the figure caption of Fig. 6e.

MINOR POINTS

1. Abstract: the two strategies are not described. That part does not read well.
 - This was edited in the revised manuscript. The following sentence was added (p1, line 29-31): “To identify host proteins involved in viral morphogenesis, we characterize the proteome of SARS-CoV-2 virions produced from A549-ACE2 and Calu-3 cells, isolated via ultracentrifugation on sucrose cushion or by ACE-2 affinity capture.”.
2. Page 2: ongoing pandemic > this might not age well (hopefully). I recommend removing the ‘ongoing’.
 - This was changed accordingly
3. The downregulation of G3BP1 is not evident in the KD of G3BP2.
 - We apologize but we are uncertain to which figure the reviewer is referring to. G3BP2 KD indeed increases G3BP1 level, making G3BP1 KD less efficient in G3BP2 KD cells compared to parental cells (Supplementary Fig 4a). However, in Fig 4c, 6d and Supplementary 4a, the global levels of both G3BP1 and G3BP2 proteins were clearly downregulated.
4. Figure 6A. Improve labelling. It is not evident how upper and bottom panels differ without reading the legend.
 - The figure has been updated to include mock-infected panels for control of N and S antibody staining performed in parallel to the experiment in SARS-CoV-2 infected cells as requested by the reviewer. The labelling has been improved as requested.

Reviewer #2 (Remarks to the Author):

Overview:

Murigneux et al. have performed a very interesting and informative study, that includes appropriate controls, sound methodology, and comprehensive analyses with orthogonal techniques that provide new knowledge in the field. This work identified 175 host proteins associated with SARS-2 virions using distinct methodologies, and focussed efforts on

characterizing how two RNA binding proteins (G3BP1 and G3BP2), which normally are found in stress granules, can favor virus assembly and act in a proviral manner. This work is novel and has not been reported elsewhere, reveals new molecular mechanisms in virus assembly, and highlights the interaction of viral N proteins and G3BP proteins as a new potential antiviral target. Overall this manuscript was well-written (it was a pleasure to read!) and provides new knowledge to the field with broad applications that are also very timely in the current stage of the ongoing COVID-19 pandemic.

Comments to be addressed by the authors:

1. Line 194-195: I'm confused about the language here. Were these tetraspanins not detected in any of the analyses at all, or were they only absent from your SARS-CoV-2 network analyses? I would be surprised that exosomes were completely eliminated from your preparations, as there are exosomes that would be similar size to your virus particles that would co-purify with your ultracentrifugation and filtration protocols. Also, many exosome markers can also be present on virion surfaces, as they are not exclusive to only exosomes, particularly for viruses like SARS-2 that bud. Please clarify this in the writing.

- We apologize for the confusion: Tetraspanins such as CD151, TSPAN14, TSPAN6, CD63, CD81 and CD9 were detected in preparations from mock- and SARS-CoV-2-infected A549-ACE2 and Calu-3 cells. According to the quantitative analysis of LC-MS data following reviewer's 1 recommendations, CD151, TSPAN14, TSPAN6 and CD9 were enriched in virions produced from A549 cells, but not from Calu-3 cells. However, classical exosome markers such as CD63, CD81 and TSG101 were not enriched in the preparations from infected cells (Supplementary Dataset 1) and this was also confirmed by western-blotting (Supplementary Figs. 1a, 1c and 3b). Thus, these proteins were not included in the network analysis.
- For clarity, we moved the sentence and in the Result section (p8, line 175-179) now write:
"Of note, tetraspanins were identified in both mock- and infected A549-ACE2 and Calu-3 derived samples. However, classical exosome markers, CD63, CD81 and TSG101, were not enriched in virion preparations, suggesting limited exosome contaminants among identified virion-associated factors"

2. Line 198 – please change 'quasi totality' to a more definitive/descriptive word choice.

- p8, line 180-183: This was changed to: "Cross-referencing our data with a previously published transcriptomic study performed in nasopharyngeal swab and lung biopsies of patients infected with SARS-CoV-2⁵¹ further indicated that **97.7% (348/356)** of SARS-CoV-2 virion associated factors were detected in natural cellular targets of infection (Supplementary Fig. 2b)".

3. Line 218-220: Please re-word for clarity. I had trouble understanding what was meant here.

- We apologized for any lack of clarity. We have now modified the paragraph (p9, line 206-211):
"In particular, 27 factors belonged to a core of 88 SARS-CoV-2 RBPs identified in common in at least three different studies. These proteins likely represent a specific subset of the 503 SARS-CoV-2 RBPs identified in the literature. For example, viral proteins otherwise strongly associated with gRNA during replication, such as nsp3, nsp6 or

nsp9^{29,31,33,54}, were not detected in virions (Fig. 1b, Supplementary Figs. 1a and 1c, Supplementary Dataset 1).”.

4. Line 224-229 and Suppl Fig 1C: More details of this analysis would make the comparison of your data on SARS-2 to other viruses more useful. For example, which 21 enveloped viruses were studied here? What methods were used in the Dicker et al and Gale et al. papers? It's surprising that common surface proteins are not shown in Fig 1C, particularly since these are enveloped viruses, including for example, tetraspanins. Can you explain this?

- Reviewer 3 requested to shorten this paragraph. For more details, the names of the 20 enveloped viruses used for comparison with SARS-CoV-2 virions are now specified in the caption of Supplementary Fig. 2C. However, a wide variety of methods were used to isolate virions in these different studies and could not be detailed in the present study.
- Following suggestions of reviewer 1, our LC-MS data were re-analyzed using a quantitative analysis to identify enriched factors in virion preparations rather than factors exclusively present. Additional virion associated factors were identified, including several Tetraspanins; CD151, TSPAN14, TSPAN6 and CD9. They are now included in the Fig 1c, 2a, 2b and Supplementary Fig 2c (see also Supplementary Dataset 1).

5. Fig 3e immunoblots: Please provide more details of how these expts were performed to control for loading and comparisons of protein across samples. How much protein was loaded in cell lysate lanes vs virion lanes? How was the virion input normalized? Similarly, how much total protein was loaded in Fig 3F, and how was this controlled/normalized for flow through vs affinity-captured virus? What was the total virus 'input' in lane 1? Please provide more of these details for the reader to appreciate how much protein (virus lysate) is being analyzed and detected.

- We apologize for omitting this information.
- Since protein levels in virion-associated fractions were below the Bradford assay's quantification limit, identical volumes were loaded in all experiments instead of identical protein levels. This is now stated in the caption of the corresponding figures.
- Figs. 1b, 3e and 3g, as well as Supplementary Figs 1a and 1c: 0.25% of cell-associated fraction was loaded (around 20 μ g of proteins according to Bradford quantifications) and 2% of virion-associated fraction was loaded.
- Figure 3f: The input material used corresponds to the lysate of virions isolated through ultracentrifugation on a sucrose cushion. This clarification has been added to the figure's legend. Following ACE2 affinity capture, 4/5 of the material was allocated for mass spectrometry analysis, while the remaining 1/5 was preserved for subsequent western blot and RT-qPCR analyses. In the case of Western blot analyses, 1.25% of the total input, flow-through (FT), and ACE2 affinity capture fractions were loaded for analysis.

- All the above information has been added to the captions of the corresponding figures.

6. Lines 301-302 & Fig 4A: why did KD of YBX1 and YBX3 result in increased viral RNA? Was this expected? Can you comment on what is different about these proteins vs the others that seem to have the opposite effect (proviral)?

- The data of Fig. 4a suggest that while IGF2BP1, PABPC1 and G3BP1/2 are proviral factors in the context of SARS-CoV-2, YBX1 (and possibly YBX3 although the effect is not statistically significant) has an antiviral function. We edited the sentence regarding YBX1 (p12, line 277-279) to: “By contrast, KD of YBX3 and IGF2BP3 had no impact, and KD of YBX1 increased the level of virion-associated gRNA by 3.1-fold, **suggesting an antiviral role of this factor in the context of SARS-CoV-2.**”

Our siRNA screen primarily aimed to investigate whether well-established Stress Granule constituents participate in SARS-CoV-2 replication through their association with Stress Granules. The presence of both pro-viral and antiviral SG factors suggests a more complex relationship (p12, line 279-281). As discussed with reviewer 3, it's not surprising that some factors associated with virions are pro-viral, some are antiviral, and others have no effect, as similar observations have been reported in the literature for virions of other viruses.

- Regarding YBX1 and YBX3, both are cold shock domain proteins with overlapping functions that are known for their DNA and RNA binding capabilities. YBX1 is more characterized than YBX3. It plays roles in transcriptional and translational regulation, DNA repair as well as pre-mRNA splicing, mRNA transport and mRNA decay (Liabin et al., 2014, PMID: 24217978). Moreover, it has been implicated in the replication of numerous viruses, sometimes as a pro-viral factor and other times as an antiviral factor (Weydert et al., PLOS ONE, 2016). It was recently reported to interact with Dengue Nucleocapsid and mediate both virion assembly and release (Diosa-Toro, MBio 2022). In the context of SARS-CoV-2 replication, two independent screens have suggested that YBX-1 may be a proviral factor (Zhang et al., Cell Research, 2022; Labeau et al., Cell Reports, 2022), potentially involved in late-stage replication (Labeau et al., Cell Reports, 2022). However, these screens were conducted in different cell lines, with different output and at different infection time points than our study. Further investigation into the role of YBX-1 in SARS-CoV-2 replication, would be an intriguing avenue for future research. At this stage, we consider the data too preliminary, and more in-depth studies are needed to provide a comprehensive understanding of the multifaceted role of this protein in SARS-CoV-2 replication.

7. Line 318: why do the authors choose to refer to the virus-associated N protein as ‘extracellular’ levels in the main text (as on line 318), but ‘virions’ in the figure panel (as in Fig 4D). I would think it should be harmonized, and more accurately described a cell-associated or virion-associated.

- We thank the reviewer for the comment and edited the manuscript accordingly.

8. Fig 4C: Again, more details on how this assay was performed are needed to enable accurate interpretation of the results. What was the cell input and viral input (amount of

protein)? how was input controlled and where/what is the loading control for virion lanes? What protein measurement in viruses was used to normalize the observed reduction of virus-associated N protein (and S protein) shown in the blots and plotted in the bar graphs for 4C?

- We apologize again for not providing sufficient information. The Bradford assay of the virion-associated fraction was below the detection limit, so we could not have a loading control for the virions lane. Therefore, we used identical volumes of ultracentrifuged virion-associated fractions when loading CTL and G3BP1/G3BP2 KD samples.
- Fig 4c: To enable comparison between virion and cell associated fractions and to calculate the release index, identical volumes of cell-associated fractions were also loaded without further protein level normalization. 1.25% of cell-associated fraction (approximately 20 μ g of protein) and 25% of virion-associated fraction were loaded for western-blot analysis in each experiment. Reassuringly, GAPDH levels were equivalent between virion and cell-associated fraction, confirming that G3BP1/2 downregulation does not affect cell growth as compared to control. This information is now added in Figure caption.

9. The legend for Fig 7 should indicate what the DMV and SMV are for clarity to a non-specialist. Or if space permits, you could even just label as double/single membrane vesicle directly on the image?

- The image was edited accordingly

10. A recent paper by S.M Dolliver et al 2022 PLoS Pathogens (PMID 36534661) published some interesting work on SG formation in SARS-2 infection, with some affirming findings (absence of SG formation) and some conflicting findings (sharp reduction in G3BP in infected cells, and antiviral functions of G3BP1) to your results herein. It would be important for the authors update the text to include a discussion of this. For example, why have others observed the reduction in G3BP1 in SARS-2 infected cells but not in your study? Line 287 should also be updated to include this work and reference.

- We thank the reviewer for highlighting this study. Indeed, Dolliver et al. observed a reduction of G3BP1 at the RNA and protein level in SARS-CoV-2 infected 293A-ACE2 cells at 24hpi, but not a reduction of G3BP2. Importantly, they further suggest that the decrease in G3BP1 mRNA is due to a general cellular mRNA depletion mediated by Nsp1, a major host shutoff factor of SARS-CoV-2, rather than a targeted degradation of G3BP1.
- We also performed a kinetic of G3BP1 expression during a time course of infection in A549-ACE2 cells (MOI 0.1). A slight decrease may be visible at 48 hpi (see figure below), but was not visible either by qPCR, or by Western-blot at 10 hpi a time point when most of our experiments were performed. Importantly, Dolliver et al used a MOI 1 for 48h, which in our hands in A549 cells induces a high cytopathic effect at 48 hpi. They further studied the impact of G3BP1-eGFP overexpression on HCoV-OC43, but not on SARS-CoV-2.

Contradictory effects were observed regarding the role of G3BP proteins in SARS-CoV-2 replication. Whereas Zheng et al and Liu et al suggested an antiviral function of SARS-CoV-2, Ciccocanti et al and Kruse et al suggested a proviral function. This point is discussed with reviewer 1.

Overall, very nice work!

Kind regards,

Dr. Christina Guzzo
University of Toronto Scarborough

Reviewer #3 (Remarks to the Author):

The authors present a well-structured and controlled study in which they reveal the previously unidentified host factor content of SARS-CoV-2 particles. While this is certainly a resource for the field, it appears that the inclusion of host factors in particles is somewhat arbitrary, as other candidates besides G3BP1/2 have no or opposite impact on viral replication.

Furthermore, the key question of the importance of G3BP1/2 in particles is not addressed. The manuscript focuses on the cellular role of G3BP1/2 during viral infection, confirming their previously described interaction with N, and proposing a new proviral function. The model for the role of G3BP1/2 in particle assembly is very interesting but the conclusions remain preliminary at this stage and require additional ultrastructural evidence to be supported (see my specific comment below).

- As discussed with reviewer 1, we agree that cellular factors associated with the virus can be passively incorporated, likely due to their being highly abundant, and are not necessarily functional. This point is now clearly highlighted in the Discussion (p17, line 402-415). For this reason, subsequent analyses aimed to highlight promising candidates based on their known interactions with viral structural proteins and/or RNA, their interactions with other cellular factors, and their incorporation in other viruses in the literature. As discussed also with Reviewer 1, considering the degree of enrichment of factors in virions rather than their Intensity, may be a better way to discriminate factors that are selectively recruited to the virions. Nevertheless, validating the functional roles of individual factors through subsequent experiments to understand their association with the virion is necessary.

- Various proteins are actively incorporated into virions, and this can occur at different stages of the viral life cycle. Some proteins become part of virions because of their role in morphogenesis or budding in the late stages of viral replication, while others are necessary in the early stages of target cell infection. The former can affect level of virion production, while the latter can have an impact on virion infectivity. These virion-associated proteins can also have either pro-viral or antiviral functions. For example, cellular proteins like TSG101 and Alix are involved in viral budding and are consequently incorporated into HIV virions (Garrus JE et al., Cell 2001; Strack B et al., Cell 2003). On the other hand, APOBEC3G is incorporated into viral particles and exerts an antiviral function during reverse transcription in newly infected cells (Sheehy et al., Nature Medicine 2003). In this context, it's not surprising that some of the virion-associated factors in this study have antiviral functions (e.g., YBX1), while others promote viral replication (e.g., G3BP1, G3BP2, IGF2BP1, and PABPC1), and some may not have a discernible function (e.g., YBX3 and IGF2BP3) (Fig 4a and p12, line 274-280).
 - Numerous screens have been conducted to investigate the interaction between host cell factors and SARS-CoV-2 during its replication. However, none of these studies specifically examined the late stages of replication. While we agree that our study likely identified factors that play a role for the virion in subsequent stages of infection, our primary goal was to identify factors associated with virions during their morphogenesis. In this context, our study reveals that KD/KO of G3BP1 and G3BP2 in producer cells results in approximately a two-fold reduction in virion production (see Fig. 4b-4d). Following the reviewer's comment, we also measured the infectivity of produced virion. This is included Fig. 4e and demonstrates that the virions produced are approximately two times less infectious. Overall, this indicates that G3BP1 and G3BP2 have a pro-viral function in both virion production and their ability to infect new cells.
- A549-ACE2 cells are a valuable tool but show very slow particle release kinetics compared to the Calu-3 human airway epithelial cells, which in addition also express TMPRSS2 that is important for S maturation. It is therefore important to demonstrate that the incorporation of G3BP1 and G3BP2 into particles and their effect on particle assembly is not cell type dependent.
- We have now included an MS analysis of virions produced from Calu-3 cells and confirmed the association of G3BP1 and G3BP2 and other factors of interest with virions. Their association with virion produced from Calu-3 cells was also confirmed by western-blotting analysis.
- Is the silencing of G3BP1/2 also proviral in other relevant cell lines, e.g. Calu-3 cells?
- The effect of G3BP1/2 KD was assessed in Calu-3 cells and we confirmed a proviral effect of G3BP proteins in the production of virions and their infectivity (Figs 4b-4e).
- The presence of G3BP1/2 into purified virus particles should be validated by immunogold labeling along with S or N.
- We acknowledge that formal demonstration of their incorporation inside virions was not achieved. Immunogold labeling of a viral protein that is highly abundant within

virions is already a challenging task due to the thin section and limited antigen accessibility. Given the much lower levels of incorporated cellular proteins compared to viral structural proteins (to provide an approximate magnitude, the intensity signal in LC-MS for G3BP1 is roughly 1/20 to 1/40 of that observed for S or N), immunolabeling of G3BP1, would prove extremely challenging, even when employing cryoelectron microscopy techniques.

- However, our study reveals that G3BP1 and G3BP2 are among the most enriched host factors in virion preparations from infected cells compared to non-infected cells, with G3BP1 showing approximately a 50-fold enrichment as compared to preparation from non-infected cells (Figs. 1f and 1g). Importantly, G3BP proteins could not be detected in preparations from non-infected cells, suggesting that they are not usually secreted by cells in extracellular vesicles (Supplementary Dataset 1-. In addition, G3BP proteins have never been identified as virion-associated factors in 20 other viruses, strongly suggesting a specific association of G3BP with SARS-CoV-2 virions (Supplementary Fig 2c).
 - The detection of G3BP proteins by mass spectrometry was further validated through western blot analyses in virions produced from both A549-ACE2 and Calu-3 cells, confirming that their presence is not dependent on cell type (Fig. 3e). They were also detected with virions isolated through ACE-2 affinity capture, indicating their association with virions containing spike proteins on their surface (Fig. 3f).
 - Additionally, we were able to co-immunoprecipitate G3BP1 with the N protein from isolated virions, strongly suggesting its incorporation into the viral particle (Fig 6e).
 - To further validate the incorporation of G3BP1 and G3BP2 within virions, isolated virions were thus treated with subtilisin A, followed by further isolation by ultracentrifugation on a sucrose cushion (as suggested by reviewer 1). The protein content was then analyzed by western blotting. Notably, while the N-terminal part of the Spike protein, located outside of the virion, was digested by subtilisin A, G3BP1 and G3BP2 remained intact, providing strong evidence that they are protected within the viral particles. These results are presented in Figure 3g.
- The bioinformatics analysis comparing the interactions of the virion-associated host candidates identified in this study with other host proteins identified from cells or in different viruses (Fig. 2b and Supp. Fig. 1c) is interesting but does not add much. It could be shortened.
- We attempted to condense this paragraph, while adding the supplementary information requested by reviewer 2 in the legend of the corresponding supplementary Fig. 2.
- Additional staining for ORF7a and for nsp3, as a viral protein that is not incorporated into the particles, would strengthen the results regarding the quality of the purified fraction.
- As requested, we conducted additional staining for ORF7a and nsp3. We confirmed the presence of ORF7a in virion-associated fractions from both A549 and Calu-3 cells using ultracentrifugation on a sucrose cushion followed by ultrafiltration and using ACE2 affinity capture (Fig. 1b, 3b, and Supplementary Fig. 1a, 1c, and 3b). In contrast, nsp3, a highly abundant viral protein produced within the cells, was detected in the cell lysates (Supplementary Fig. 1c), but was never detected in isolated virions

across the different experiments (Fig. 1b, 3b, and Supplementary Fig. 1a, 1c, and 3b).

• In general, input/cell lysates and purified fractions should be loaded onto the same SDS-PAGE gel (Fig 3e, 4c, Supp Fig 2a, Supp Fig 3c). On the same line, the panels for G3BP1, G3BP2, and GAPDH staining are absent.

- Except for Supplementary Fig 4g, which could not be re-loaded, in all other experiments, the input/cell-associated fraction and virion-isolated fractions were loaded onto the same gels and transferred onto identical membranes. However, in certain instances, the band intensities were considerably stronger in the cell lysate/inputs compared to the virion-associated fraction. Consequently, we needed to cut the membrane to facilitate the accurate detection and quantification of the specific bands of interest. Nevertheless, the levels of proteins were compared between G3BP1/2 KO and CTL samples either within cell-associated or virion associated fractions that were developed simultaneously. The release index was calculated as the ratio of virion-associated over cell-associated.
- Fig. 3e and Supplementary Fig. 2a from our previous version of the manuscript now appear as a single figure, Fig. 3e, in the revised version. These Western blots did not require band quantification. Figure 3e now displays the image of the uncut membranes.
- Supplementary Fig. 3c from our previous version of the manuscript now appears as Supplementary Fig. 4g in the revised version. Band intensities in Fig 4c and Fig 4g required quantifications. Due to differences in signal intensities between the input and the isolated virions, we had to cut the membrane following primary and secondary antibody incubations and developed the cell-associated and virion-associated fraction with appropriate time exposure to allow subsequent quantification of band intensities. Nevertheless, the levels of proteins were compared between G3BP1/2 KD and CTL samples either within cell-associated or virion associated fractions that were developed simultaneously. The release index was calculated as the ratio of virion-associated over cell-associated.
- Fig. 4c and Supplementary Fig 4g: The panel for G3BP1, G3BP2 and GAPDH staining was added as requested.

• The authors mention the absence of exosome contamination. Validation of representative marker molecules in virion preparations by Western blot would support this point.

- As discussed with reviewer 2, we detected the presence of exosomal proteins by MS in viral preparations. However, typical markers such as CD63 and CD81 did not exhibit enrichment compared to preparations from non-infected cells. This suggests limited contaminants among the proteins we have identified as virion-associated. We have appropriately rephrased this statement (p8, line 175-179)
- In addition, as requested we have further confirmed by western-blot the absence of CD81, CD63 and CD9 in viral preparation from both A549-ACE2 and Calu-3 cells (Supplementary Fig 1a, 1c and 3b).

• Virus preparations shown in Fig.1b and in Fig.3b, e and f differ regarding to the levels of S2.

Does ACE2 affinity capture enrich particles carrying uncleaved S? What is the infectivity level (TCID50) of these particles compared to those passed through the sucrose cushion?

- ACE2 receptor recognizes the N-terminal part of the Spike protein, and as a result, ACE2 affinity capture enables the capture of virions with uncleaved Spike (S0) protein.
- In our original submission (Fig. 1b), we used the SARS-CoV-2 spike protein S1-NTD antibody from Cell Signaling, which was indicated as being able to detect both S0 and S1 domains, and not S2 as previously stated. We sincerely apologize for any confusion caused. Unfortunately, this antibody proved to be insufficiently sensitive and could not be reliably reused across multiple experiments.

As a result, for the remainder of our study, we switched to using the SARS-CoV-2 spike antibody 11A9 from Genetex. This antibody detects both the S0 and S2 domains of the spike protein, with increased sensitivity and stability over repeated uses. To ensure clarity and consistency when comparing spike profiles across experiments, we reloaded and blotted samples from Fig. 1b using the Genetex antibody. The updated gel version is now presented in Fig. 1b and Supplementary Fig. 1a.

- Unfortunately, we could not assess the infectivity of the viral particles isolated from ACE2-affinity capture, as virions were eluted using denaturing conditions.
- The effect observed upon depletion of G3BP1/2 is moderate, on average two folds and is not reported in TCID50/ml units.
 - The 2-fold decrease in virion release was observed after 10 hpi, ie during a single round of replication. A decrease of 5.7-fold in virion production assessed by gRNA quantification in the supernatant was observed after 48hpi (Fig 4a).
 - The 2-fold phenotype observed in a single round of replication is consistent across different system (siRNA and CRISP-CAS9 and across several cell lines (A549 and Calu-3 cells) (Fig 4 and Supplementary Fig 4). Moreover, overexpression of G3BP2-HA rescued the effect of the downregulation, confirming the specificity of this effect (Fig 4f).
 - The effect on the infectivity is reported in TCID50/mL in Figure 4d. We agree that the labeling was confusing and it has been modified accordingly.

Viability assays in the presence and absence of viral infection should be provided for the siRNA and Cas9 transient experiments to exclude an artifactual effect of the method.

- Viability assays were conducted in parallel to the experiments depicted in Figure 4, both in the presence and absence of viral infection. These assays revealed that G3BP1/2 KD had no impact on cell viability. The corresponding data have been added into Supplementary Figure 4c.
- The method used to deplete G3BP1 and G3BP2 appears to impact intracellular gRNA levels. In transient Cas9 assays (Fig. 3b), levels are reduced by 25% while unaffected by silencing. This undermines the main argument that G3BP1 does not impact replication but assembly.

As the decrease in level of cell-associated gRNA in G3BP1/2 is inconsistent from one method to the other and from one cell line to the other, we believe that this effect is not reliable. It is noteworthy that the level of virions produced in the supernatant of cells previously transduced with CRISPR-CAS9 and guide RNA, whether they were CTL or G3BP gRNA, was lower than in siRNA transfected cells.

- Nevertheless, the decrease in the level of RNA associated with the virion fraction in G3BP1/2 KO as compared to CTL, still resulted in a 50% decrease in the release index at the RNA level (unnormalized virion-associated level/unnormalized cell-associated level) in G3BP1/2 KD as compared to control conditions. We admit that the way the figure was presented in the previous version of our manuscript (with the relative intracellular and relative extracellular panels) was confusing. We have now improved the Figures by presenting the cell-associated, virion-associated and Release index (Fig 4b-e and Supplementary Fig 4 F-h).
- In addition, and importantly, even though the cell-associated viral N sgRNA level was decreased by 25% in KO cells, this was not reflected in the cell-associated N protein level. Moreover, the 2-fold decrease in the production of virions was confirmed with N protein and in TCID50.

• A549 stable G3BP1/2 double knockout cells seem to be viable (<https://www.biorxiv.org/content/10.1101/2021.04.26.441141v2>). Such cells would help clarifying the effects.

- We thank the reviewer for his valuable suggestion. We received the A549 control and Δ G3BP1/2 clonal cells from the laboratory of Sun Hur. In line with our findings from KD and transient KO experiments, these cells allowed us to corroborate a 1.6-fold reduction in the release index of virion assessed at the RNA level (Fig. 4f). Furthermore, these cells allowed us to demonstrate that the overexpression of G3BP2-HA in a cellular background lacking G3BP1 and G3BP2 successfully restored the release of virions.

• The fact that binding between N and G3BP1/2 depends on an interaction with RNA in cell extracts is interesting. How were the virion samples treated with RNase? Were virions ruptured first? This information is missing from the method section.

- The virions were lysed before RNase treatment and we verified by RT-qPCR that the viral gRNA was degraded in cell and virion extracts. No traces of RNA were detected in either cell or virion lysates after RNase treatment (see answer to referee 1).
- We apologize for the missing information in the Method section and have modified the section accordingly (p 26, line 626-634).

• Fig.6d, representative Western blot image: the level of S in the total lysate is visibly reduced. This is misleading. In addition, total lysates and purified fractions should be analyzed on the same gel.

- As discussed with reviewer 1, for technical reasons, this experiment was conducted at 24 h post-infection (hpi) after several rounds of viral replication. The reduction in

the levels of both S and N proteins in the total cell lysate is expected and reflected the decrease of SARS-CoV-2 replication after multiple rounds of replication in G3BP1/2 KD cells. However, in the membrane fraction, the relative level of N to S was reduced by 2-fold, as compared to the relative level of N to S in the total cell lysate. This suggests that N is recruited less efficiently to the virion assembly site.

- Both purified fractions and total cell lysates were loaded on the same gel and transferred onto the same membrane (see raw data). However, due to differences in signal intensities between the cell lysates and the purified fractions for S, we needed to cut the membrane after primary and secondary antibody incubations. This allowed us to develop the cell-associated and virion-associated fractions with appropriate time exposure for subsequent quantification of band intensities of S. It's noteworthy that the quantified ratios are calculated within total cell lysates and within membrane fractions, ensuring that our results are not biased.
- Supp Fig 4 shows the degree of colocalization between G3BP1 and N in infected cells. This type of analysis is not provided for the expansion microscopy. What is the degree of colocalization between N and S and N and M in ctrl vs. G3BP1/2 kd cells? This information would provide evidence to support the model.
- Doing colocalization analysis on the expansion microscopy images was not possible, for two main reasons: (i) The Pearson coefficient for colocalization relies on the fluorescence intensity. After expansion, not all probes survive the treatment with the same efficiency. (ii) The number of cells acquired is too low to get significant amount of data.
 - As requested, we have included the Pearson correlation analysis between N and S in CTL (control) vs. G3BP1/2 KD cells in Supplementary Fig 6. However, it's important to note that, on a global scale, N does not significantly colocalize with S in CTL cells ($P = 0.2$). This makes it challenging to assess the overall impact of G3BP1/2 KD on the global colocalization of N and S. Nonetheless, the reduction in the colocalization of N and S in cytoplasmic foci, which likely represent sites of virion assembly or accumulation, was notably pronounced in Figure 6c, showing a 2-fold decrease.

Can G3BP1/2 interact directly with S?

- Systematic characterization of interaction S with host proteins has been performed by several laboratories and never detected an interaction with S. (Davies et al, ACS Infectious diseases 2020; Gordon et al, Science 2020; Stukalov, Nature 2021, see also Biogrid Project).
- Expansion microscopy is an impressive method for improving resolution. However, when it comes to visualizing differences affecting viral assembly sites, analysis at ultrastructural level is more relevant. Electron microscopy combined with immunogold labeling of S and N would support the observations and the proposed model more convincingly. This would additionally exclude the possibility that the substructures observed in expansion microscopy are in fact

intracellular accumulation of viral particles.

- As requested, we performed pre-embedding immunogold labeling of G3BP1 proteins in A549-ACE2 cells infected with SARS-CoV-2 for 10 h. In this process, cells were fixed, permeabilized, and subjected to incubation with a rabbit anti-G3BP1 antibody, followed by incubation with a 0.8 nm Nanogold secondary antibody targeting rabbit antigens, and subsequent silver enhancement. Samples were then embedded and analyzed using electron microscopy. This protocol resulted in the formation of approximately 5 nm-sized beads. The size of the beads varied depending on the silver enhancement and the bead's localization within the observed section. As illustrated in the left figure below, in approximately one-third of the cells, we observed the emergence of "bags" of dark vesicles corresponding to SARS-CoV-2 virions, that are characteristic of SARS-CoV-2 infection. These structures were only observed in infected cells. Upon closer examination, we observed a typical G3BP1 immunogold staining after a silver enhancement procedure, that accumulates around and potentially inside these virions (left image). Nevertheless, despite multiple attempts, we were unable to obtain images of the required quality for publication.

- However, while we cannot distinguish definitively between sites of virion assembly and sites of virion accumulation through expansion microscopy, our findings clearly indicate that G3BP1/2 KD impact late stages of virion assembly/release. This includes a two-fold reduction in the recruitment of N to S-associated membranes (Fig 6d) , a corresponding two-fold decrease in the formation of ERGIC derived structures of viral structural proteins and gRNA accumulation (Fig 6c), and a two-fold reduction in virion production (Fig 4b-4d) and virion infectivity (Fig 4e). Moreover, overexpression of G3BP2-HA can rescue the release of virions in G3BP1/2 KO cells (Fig 4f). Together, these findings strongly support a model in which G3BP proteins play a role in the production and/or release of virions into the cell supernatant.

REVIEWERS' COMMENTS

Reviewer #1 (Remarks to the Author):

I would like to thank the authors for taking my comments on board and implementing the most important suggestions to the manuscript. I think the paper has improved with the modifications suggested by the different reviewers.

In other instances where authors have not added additional data, they have been able to convey their reasoning.

I do not have further comments.

Reviewer #2 (Remarks to the Author):

The authors have adequately responded to all of my queries, and I appreciate the detailed and thoughtful responses.

Reviewer #3 (Remarks to the Author):

Murigneux and co-authors have provided thorough responses to the questions and suggestions raised by every reviewer. Through this additional effort, all my remaining concerns have been addressed, resulting in a significant improvement in the overall quality of the manuscript. I believe the results of this research will be of great value to the field.